# Impacts of Climate Change on Work Health and Safety in Australia: A Scoping Literature Review

**DOI:** 10.3390/ijerph20217004

**Published:** 2023-10-31

**Authors:** Lucia Wuersch, Alain Neher, Frank E. Marino, Larissa Bamberry, Rodney Pope

**Affiliations:** 1School of Business, Charles Sturt University, Bathurst, NSW 2795, Australia; 2Research Group for Human Adaptation, Exercise & Health, School of Allied Health, Exercise & Sports Sciences, Charles Sturt University, Albury, NSW 2640, Australia; 3School of Business, Charles Sturt University, Albury, NSW 2640, Australia; 4School of Allied Health, Exercise & Sports Sciences, Charles Sturt University, Albury, NSW 2640, Australia

**Keywords:** climate change, work health and safety (WHS), workplace, Australia, literature review

## Abstract

This scoping review explores the extant literature on climate change impacts on Workplace Health and Safety (WHS) in Australia. It maps the coverage of climate hazards, occupations at risk, and health and socio-economic impacts with the aim of identifying climate change impacts on WHS in Australia and associated knowledge gaps. We used a scoping review approach to identify and investigate 41 scholarly works at the nexus between climate change and WHS in Australia. Thematic template analysis and the NVivo software helped us identify and structure the main themes and systematically document the analysis process. The review highlighted a research focus on the impacts on WHS of heat and extreme weather events resulting from climate change. Agriculture and construction emerged as the most examined occupations, emphasising climate-related diseases and productivity loss. Other climate-related hazards, occupations, and health and socio-economic impacts were largely overlooked in the included research literature. The analysis revealed there is scope for further research relating to climate change impacts on occupational hazards (e.g., air pollution), occupations (e.g., indoor settings at risk), worker health (e.g., injuries), and socio-economic impacts (e.g., change in social practice). Furthermore, the results highlight that the main themes (hazards, occupations, health, and productivity) are interconnected, and the impacts of climate change can be ‘cascading’, adding complexity and severity. Hence, it is important to look at WHS as a multifaceted phenomenon in a holistic way to understand the risks and support required.

## 1. Introduction

Research suggests global climate change is caused by human-made factors such as urbanisation, deforestation, population growth, and energy policies [1,2]. Global climate change, in turn, triggers a series of climate-related natural hazards, including increased ambient temperatures, air pollution, ultraviolet radiation, extreme weather events, and vector-borne diseases [1]. Globally, 90% of all major disasters are due to climate-influenced hazards [3]. In Australia, the total cost of all-natural disasters, including hail, floods, tropical cyclones, and wildfires, for the period 1967–1999 was estimated at AUD 37.8 billion, of which only about AUD 5 billion was attributable to non-climate-related disasters, such as earthquakes [4]. In 2023, the Australian Government released its *Intergenerational Report*, identifying that climate change will have a significant impact on productivity and the economy over the coming decades [5]. From 2002 to 2010, Australia experienced three severe droughts, which comprised the Millennium drought, with negative long-term impacts on agricultural productivity. As a result, between 1996 and 2014, total agricultural productivity remained about 50% behind the historical trend from 1950 to 1996 [6]. In the future, agricultural productivity and labour productivity are predicted in Australia to further decline due to climate change: by 2030, more than USD 19 billion; by 2050, more than USD 211 billion; and by 2100, more than USD 4 trillion are expected in accumulated wealth loss [3]. As a consequence of reduced work performance due to heat, the annual economic burden for Australia is about USD 655 per person across the workforce, which is about USD 6.2 billion for the entire Australian workforce [7].

Schulte and Chun’s [2] (p. 547) review of scholarly literature published from 1988 to 2008 emphasised that “[t]here is a substantial body of literature on the association between weather disasters and death, injury, communicable diseases, malnutrition, famine, and mental health disorders”. In particular, ‘increased ambient temperatures’ have received much attention in scholarly works in the past years in Australia (for example, references [7,8,9,10]). It has been identified that the number of days per year with temperatures above 35 °C has been steadily increasing, with this assumed to be a continuing trend [11,12]. Indeed, compared to 1957, there have been, on average, yearly 12 more days above 35 °C in Australia in 2015 [13]. Looking ahead to 2041–2080, Hall et al. [8] examined the likely impacts of climate change on Wet-Bulb Globe Temperatures (WBGT) in summer and winter across Australia to gauge the extent to which outdoor physical work capacity and risk of heat injury may be affected. Based on that work, a WBGT increase of 2–3 °C is predicted in the tropics and mid-latitudes in the 21st century, and workers are at increased risk of productivity loss and heat illness [14]. In Australia, heat is responsible for over 1100 deaths per year, which presents the highest risk among natural hazards and a risk exceeding the annual road toll [15].

Schulte et al.’s [1] framework identifies how climate change could affect workers, workplaces, occupational morbidity, mortality, and injury and has been used in studies assessing climate change impacts on the health of outdoor and indoor workers (for example, references [16,17,18,19]). Others have noted that global climate change represents a crisis for the quality of work, with significant implications for Work Health and Safety (WHS) and workers’ well-being and resilience [20]. This is confirmed by Schulte et al. [1] (p. 858), who stated that “the recent scientific literature indicates that climate change is presenting and will continue to present occupational safety and health hazards to workers”. In particular, existing research recognises heat stress as a major risk factor for WHS and finds that almost half of the global population and over one billion workers have experienced heat stress through exposure to repeated high-heat episodes [21]. As a result, about a third of all exposed workers suffer from heat-related health issues [21]. Indeed, higher temperatures significantly increase workplace injuries and are estimated to cause around 20,000 yearly injuries in California (United States) alone [22]. In Australia, the 2009 heat waves led to various occupational health deaths [23]. Also, a narrative literature review based on 32 works selected from 745 initially identified studies assessed the impacts of climate change on workers’ health and productivity from 2002 to 2019 [24]. The review concluded that “substantial numbers of workers are experiencing the health effects of elevated temperatures, in combination with changes in precipitation patterns, climate extremes, and the effects of air pollution, which have a potential impact on their safety and well-being” [24] (p. 1).

The shift from a focus on climate change impacts on ecosystems and environmental change only occurred after 2000, when research started to investigate climate change impacts on health [25]. However, the extent of health impacts caused by climate-related hazards varies among different occupations and industries. For example, outdoor activities are riskier for heat stress than activities in air-conditioned indoor spaces [11]. Furthermore, climate change as a consequence of the capitalist mode of production is arguably anthropological and impacts not only workers but the sustainability of entire organisations. In this way, climate change “is destabilising for both the performance of labour on particular worksites and the process of accumulation and social reproduction more generally” [17] (p. 12). While this interference endangers profitability, it may present opportunities to change production processes by asking, “How are climate issues already embedded in the labour process and our working lives?”. Hence, strategies to address climate change impacts need to be embedded within internal organisational processes.

Extant literature reviews [10,24,26,27,28] have focused on heat impacts on WHS in Australia. However, referring to Schulte’s [1] framework, ‘increased ambient temperatures’ is just one climate-related hazard, among others. Little is known about other climate-related impacts, such as extreme weather, vector-borne diseases, air pollution, and ultraviolet radiation, on WHS in Australia. Such information would be helpful, as in recent years, and partly alongside a global pandemic crisis, some parts of Australia have experienced a series of extreme climate events [29]—starting with a drought in 2019, followed by the Black Summer fires in 2019/20, and then a series of severe floods in 2020–2022. In particular, in regional New South Wales (NSW), Australia, some communities have been affected by several climate-related hazards in a cascading manner [30].

Furthermore, research on the impacts of climate-related hazards on WHS currently concentrates on agriculture and construction [21,27] and other industries where workers perform their labour outdoors. There has been limited research on industries such as healthcare and social assistance, education, financial and insurance services (an exception is [19]), or other industries where work is predominantly performed indoors. Therefore, the impacts of climate-related hazards on many occupations and industries have received little scholarly attention. Much of the existing literature on the impacts of climate change on WHS in specific Australian industries or occupations has explored this issue in aggregate terms at a national or broader geographic level, with little exploration of the differences between metropolitan and regional or remote settings. However, Bi and Parton [4] emphasise that it is vital to explore the impacts of climate-related hazards in regional and remote locations, given the inequality of access to healthcare and other services in these locations. Building on this analysis, it is essential to note that, in Australia, regional labour markets are distinctively different from metropolitan labour markets [31], with a significantly narrower range of industries and a flatter occupational structure, skewed towards lower-skilled roles [32]. Outdoor work in industries such as agriculture, mining, and construction is overrepresented in regional labour markets, while the distance and remoteness of regional locations have implications for work that is predominantly undertaken indoors but requires outreach and travel, such as in the community health and disability sectors. Regional and rural populations have also been found to be older on average than metropolitan populations [31,32,33] and have a more constrained skill structure, reflecting both limited training opportunities and the regional occupational structure.

Therefore, the impacts of climate-related hazards on many professions and industries in Australia have received little scholarly attention. This study aimed to provide an overview and mapping of what is known about climate change impacts on WHS in Australia and identify key knowledge gaps. As a major contribution, this study aimed to provide a ‘catalogue’ summarising the coverage of extant literature regarding climate change impacts on WHS. Such a catalogue can be used by various WHS stakeholders, such as policymakers, as a source of information when developing WHS policies. Furthermore, our catalogue can help organisations guide climate change and WHS discussions with their stakeholders, particularly in at-risk geographical areas and occupations. Specifically, this study asked the question, *What is known from the existing literature about how climate change impacts WHS in Australia?*

## 2. Methodology

Conducting a literature review is a “more or less systematic way of collecting and synthesising previous research” [34] (p. 333). A literature review can provide an overview of disparate and interdisciplinary research areas, help synthesise research outcomes, showcase evidence on a meta-level, and uncover areas where more research is needed [34]. For example, extant literature reviews on climate change impacts on WHS have used a systematic approach [27], a narrative approach [24], a comprehensive approach [28], an epidemiological approach [10], or a global approach [26]. In this study, the research question was addressed using the well-established approach of a scoping literature review [35].

### 2.1. Approach: Using Scoping Literature Review as a Methodological Approach

Our literature review follows the methodological approach of a scoping literature review [35], one type of systematic review [36]. The systematic approach is arguably the most rigorous literature review method and uses clearly defined processes to elicit all evidence on a given topic [37,38,39]. Such detailed research leads to reliable results and minimises the effects of possible researcher bias [37,40]. According to the Joanna Briggs Institute (JBI), systematic reviews include various kinds of review methods, such as systematic reviews of experiences or meaningfulness; effectiveness; text and opinion/policy; prevalence and incidence; costs of a certain intervention, process, or procedure; aetiology and risk; mixed methods; or diagnostic test accuracy; and umbrella reviews; and scoping reviews [36]. The Preferred Reporting Items for Systematic Reviews (PRISMA) Statement was extended in 2018 to Scoping Reviews—the PRISMA-ScR [41].

Scoping reviews, also called ‘scoping studies’ [36], allow for the preliminary assessment of the potential size and scope of available research literature in a certain field and aim to identify the nature and extent of research evidence [38]. We decided to conduct a scoping review, as it uses a systematic approach and is well suited to investigate broad interdisciplinary topics and research questions [39], such as those relating to the impacts of climate change on WHS. In contrast to other literature reviews seeking to identify the most significant items in a certain research field (critical review), aiming at identifying component reviews while excluding primary studies (umbrella review), or determining searches by time constraints (rapid review), the scoping review is limited in its completeness by scope constraints [38].

Furthermore, our research aim matches one of the primary purposes of a scoping literature review, which is “[t]o identify research gaps in the existing literature” [35] (p. 21), and while “[s]pecifically designed to identify gaps in the evidence base where no research has been conducted, the study may also summarise and disseminate research findings”. As a scoping literature review does not usually assess the quality of the identified literature, the identified gaps may “not necessarily identify research gaps where the research itself is of poor quality” [35] (pp. 21–22). Therefore, our scoping review was designed to “‘map’ relevant literature in the field of interest” [35] (pp. 20–21). Our mapped catalogue of topics highlights covered and uncovered aspects (quantity) of climate change impacts on WHS in Australia and offers recommendations on where future research could be more meaningful (quality).

### 2.2. Procedures: Applying the Five Stages of Conducting a Scoping Literature Review

In conducting the scoping review, we followed the five-step process proposed by Arksey and O’Malley [35] as a methodological framework: identifying the research question; identifying relevant studies; study selection; charting the data; and collating, summarising, and reporting the results. In the first stage of *identifying the research question*, and inspired by Arksey and O’Malley [35] (p. 23), who formulated their research question starting with “What is known from the existing literature about…”, we created the research question: *What is known from the existing literature about how climate change impacts WHS in Australia?* Answering this research question would allow us to understand the current state of knowledge at the nexus between climate change and WHS in Australia and to identify where extant literature is accumulated and what areas have received little attention, forming a knowledge gap.

In the second and third stages of *identifying relevant studies* and *study selection,* we sought and confirmed literature to be included in the review. Arksey and O’Malley [35] suggest searching for relevant sources, predominately using electronic databases and existing networks and conferences, to identify what databases are generally used and key journals in a specific research field. Similar to Lundgren et al. [42], who investigated the effects of heat stress due to climate change on WHS, and Moda et al. [24], who explored climate change impacts on outdoor workers’ safety, we used the PubMed database. Furthermore, based on expert advice, we also utilised the databases Cinahl Plus/EBSCO Host, Emcare/Ovid, Medline ALL, and Health Collection (Informit). From networks and a regional conference, we gathered some advice on key articles that were obtained. The reference lists of these seminal sources were hand-searched to identify additional studies, and search terms used in the reviewed sample were considered. For example, a previous literature review [42] on the effects of climate change and associated heat stress on WHS used the search terms ‘heat stress’, ‘occupational heat exposure’, ‘occupational heat stress’, ‘occupational heat strain’, ‘heat in/at the workplace’, ‘work in the heat’, and ‘occupational heat stress AND climate change’.

Based on our research question and in order to identify research evidence relating to all types of climate change hazards [1,2], we used the search term ‘climate change’ instead of ‘heat*’. Furthermore, we combined the main concepts from our research question to formulate a search string that combined ‘climate change’ using the Boolean operator ‘AND’ with ‘workplace’, AND ‘health’, AND ‘Australia’. The search was constructed to identify these search terms if they were located in a relevant source’s title, abstract, or body.

Our inclusion and exclusion criteria were informed by Snyder [39], who categorised literature reviews into three categories: systematic, semi-systematic, and integrative. Our scoping study corresponds to Snyder’s [39] ‘semi-systematic’ category, allowing contributors to identify ‘themes in the literature’ and put up a ‘research agenda’, which corresponds to the purpose of the review. However, Snyder [39] (p. 334) also notes that “there are many other forms of literature reviews, and elements from different approaches are often combined”. Our review benefited from such an adaptation, involving, in addition to peer-reviewed journal articles, the integration of research reports not published in academic journals (grey literature) to incorporate evidence from various disciplinary silos, such as ‘climate change’, ‘health’, and ‘WHS’ “in which the research is disparate and interdisciplinary…to uncover areas in which more research is needed” [39] (p. 333). Consequently, we included in the scoping review mainly peer-reviewed articles (n = 38) and some other selected research reports (n = 3).

To refine our database search, ensure the selection of literature for inclusion was robust, and further validate our search strategy, we also sought and considered advice from experts in climate change and WHS research fields. Their suggestions confirmed what we identified as seminal works based on citation numbers. This way, expert advice added to the thoroughness of the review and offered new insights, thus expanding the range of included sources. We further followed the quality assessment approach of Gebayew and colleagues [43], ensuring the selected scholarly works were valuable for research or practice. Further inclusion criteria were the English language to avoid translation costs and the mention of ‘Australia’ as a significant study setting. Furthermore, the included studies had to be relevant to answering the research question, and we also included previous literature reviews if they met other inclusion criteria.

A preliminary electronic search yielded a manageable number of studies for potential inclusion. Therefore, we decided not to restrict the date range and to include both qualitative and quantitative studies. A full reading of each paper subsequently identified in the final search (116 articles in total) confirmed initial eligibility perceptions derived from preliminary scans of titles and abstracts, with particular attention to excluding articles that mentioned the key search terms without further discussing the topic of climate change impacts on WHS in Australia. Applying these eligibility criteria to the initial 116 search results led to 75 exclusions and the inclusion of 41 sources (Appendix B), which focused on a range of occupational populations. Table 1 lists the databases and other sources considered in the search, as well as the search terms employed, the number of articles initially identified from each source, and the numbers included based on eligibility during the selection process described further below. The overwhelming majority of the resulting articles investigated topics relating to ‘heat’.

Within the included literature on climate change impacts on WHS in Australia, most eligible articles were published in the last two decades, with major peaks of published articles of relevance to the topic of our review in 2015, 2018, and 2019 (Figure 1). The number of eligible articles published in each publication year (1989–2022) illustrates that most of these works were published in 2015. This finding highlights that climate change impacts on WHS in Australia have only recently become a research focus.

Highly cited works in the field of climate change impacts on WHS have generally focused on heat impacts (for example [7,9,10]; Table 2). Furthermore, high-quality journals publishing seminal works related to climate change impacts and WHS are the *International Journal of Environmental Research and Public Health* (e.g., [19,24,45,46]); *Environmental Research* (e.g., [12,26]); *Science of the Total Environment* (e.g., [47,48])*;* and *Policy and Practice in Health and Safety* (e.g., [49,50]). Table 2 lists some highly cited works on climate-change impacts, particularly heat impacts on WHS, according to citations.

### 2.3. Analysis: Creating a Final Template

Arksey and O’Malley’s [35] fourth stage describe the analysis process for the identified study sample, in *charting the data*. We used thematic template analysis [52] as a rigorous method to qualitatively analyse eligible source materials, which included 41 scholarly works. According to King and Brooks [52], thematic analysis is a technique that can be used to identify, analyse, synthesise, and report recurring themes as general patterns in semi-structured literature reviews. Template analysis [52] consists of a six-step process to identify themes discussed in the extant literature on specific topics—in this review, climate change impacts on WHS in Australia. The six steps involve familiarisation with the data, preliminary coding, clustering, producing an initial template, developing and applying the template, and writing it up. We used Saldaña’s [53] thematic coding techniques and the QSR NVivo Version 12 (NVivo V12) software application to ensure rigorous organisation of the selected sources, the systematic categorisation of the emerging themes, and a well-documented coding and analysis process. The NVivo V12 software application helped us organise the emerging themes meaningfully and retrieve coded data [53].

Eventually, a final template was developed through an iterative process by adding emerging themes, redefining existing themes, regrouping patterns, and merging similar themes. This final template (Appendix A) is a product of Arksey and O’Malley’s [35] fourth stage of *charting the data* and lists the emerging themes, organised in interrelated clusters, as a basis for the later writing. A cluster is a group of meaningfully ordered themes relating to each other, within a group, or between them [52]. The final template provides an overview of what is known about climate change impacts on WHS in Australia and some trends in the research area.

Our final template consists of four thematic clusters, embracing 62 themes and sub-themes. It formed the backbone for the last stage of the scoping literature review, which involved *collating, summarising, and reporting the results* (Section 3). Methodological guidance suggests that, in this final stage of a scoping review, not all themes need to be discussed; instead, particular themes should be selected, for example, based on their frequency or ability to answer the research question [52]. Subsequently, the four main clusters identified in our review are discussed to answer the research question, with these comprising climate-related hazards impacting WHS in Australia (Section 3.1), climate change impacts on occupations and industries in regional Australia (Section 3.2), climate change impacts on the health of Australian workers (Section 3.3), and socio-economic impacts of climate change (Section 3.4). The remaining clusters fall outside the scope of this review and may be further explored in future research.

## 3. Findings

Consistent with Arksey and O’Malley’s [35] final stage of *summarising and reporting the results* of a scoping review, here we outline the findings of the review based on the final thematic template (Appendix A) that emerged from the included studies (Appendix B). All 41 included sources reported on Australia. Additionally, Figure 2 offers a map that conceptualises the research coverage (bold) and the associated gaps in the literature per main topic area. As a visual representation of the research question, which here is *What is known from the existing literature about how climate change impacts WHS in Australia?*, our conceptual map (Figure 2) illustrates the findings regarding climate change-related hazards as defined by Schulte et al. [35] and their impacts on WHS in Australia mapped against the industries identified in regional Australia [35]. Such mapping can highlight areas of minimal research coverage and offer opportunities for further research avenues, thus representing a contribution to the theory in itself. ‘Impact factors’ (arrows) influence interconnected climate-related areas of WHS in Australia (circles). The coverage and knowledge gaps within each circle, which emerged during the analysis of the included literature using an inductive approach, are explained in the following Section 3.1, Section 3.2, Section 3.3 and Section 3.4.

### 3.1. Climate-Related Hazards Impacting WHS in Australia

The reporting of the included studies on the seven categories of climate-related hazards [1] impacting WHS in Australia was highly variable (Table 3). Most articles (n = 32) concentrated on ‘increased ambient temperatures’ and investigated climate change impacts, including heat, heat stress, and the urban heat island phenomenon. Many articles (n = 18) dealt with ‘extreme weather’, with the ratio of these focused on urban (n = 5) and rural/regional (n = 6) areas being about balanced. While sources concentrating on urban areas generally mentioned ‘extreme weather’, those focusing on rural/regional Australia specifically highlighted ‘drought’, ‘wildfires’, and ‘floods’. In contrast, there was moderate coverage of issues including ‘air pollution’, ‘ultraviolet radiation’, ‘vector-borne disease and other biological hazards’, ‘industrial transitions and emerging industries’, and ‘changes in the built environment’, where, understandably, a focus on urban areas stood out.

Table 3 highlights that whilst there is a significant body of literature that has investigated climate change impacts in Australian workplaces related to heat, heat waves, heat stress, and urban heat islands (increased ambient temperatures) and also drought, wildfires, and floods (extreme weather), the impacts of other climate-related hazards (e.g., ultraviolet radiation and air pollution) on WHS have been under-researched in the Australian context.

In addition to the framework of Schulte and Chun [2] and Schulte et al. [1], a new ‘cascading’ category of climate-related hazards emerged, which Ingham et al. [30] termed ‘cascading disaster events’. This category refers to natural events in close succession that do not allow affected individuals, organisations, and communities to fully recover from one event before the next occurs (see Section 3.1.8). The phenomenon of ‘cascading disasters’ and their impacts on WHS have a special meaning in Australia’s NSW, where a close succession of droughts, fires, floods, and the pandemic occurred over the last three years. The following sections summarise findings from the review of climate-related hazards.

#### 3.1.1. Increased Ambient Temperatures

Increased temperatures and heat-related issues were the most covered topics in the included literature, with 32 sources and over 140 text passages reporting on these impacts of climate change. This high frequency can be explained by a quote from Oppermann and colleagues [54] (p. 885), noting, “[o]ne of climate change’s most certain impacts is increasingly frequent and extreme heat”. Indeed, climate change is resulting in increases in average temperature [27], and various climate models show a warming trend across Australia [4]. Temperatures in Australia are predicted to rise by an average of 0.6–1.3 °C by 2030 [60].

Major heat waves, that is, extended periods of abnormally high temperatures, have been associated with increased public health burdens in recent years [58]. Heat stress contributors include the body’s internal heat production through muscular physical activity, ambient heat, and clothing, which affect sweat evaporation [9]. While the impact of heat on the general health of the population is well documented, the effects of heat on occupational health are relatively neglected [16]. Nevertheless, there is growing evidence of the link between heat and impaired WHS [28]. Additionally, the predicted increases in ambient temperatures accompanying climate change and the associated increases in heat stress in the workplace pose a particular challenge for WHS in Australia [10].

Hot ambient temperatures are problematic for WHS because of their link to an increased risk of occupational diseases and accidents [47]. Through high temperatures and high humidity, the human thermoregulatory system becomes impaired, and heat-related diseases can occur [27]. Because physical work activity generates intrabody heat production, exacerbated by environmental heat stress, workers in hot seasons and hot locations in Australia are especially vulnerable to heat-related health problems [27]. Extant research indicates that excess heat levels are associated with increased mortality among older people in Australia and elsewhere [27]. However, study results also confirm the link between exposure to high heat levels in Australian workplaces and health effects on workers in younger age groups [27].

Given that Australian climate change projections predict increasing temperatures, extreme weather-related heat effects will mean an increasingly high risk of heat-related illnesses and injuries for many employees, both outdoors and indoors [10]. There is more knowledge of heat-related diseases than heat-related injuries [28]. Injuries in workplace settings are clearly linked to high ambient temperatures [16], leading to decreased cognitive motor skills and heat exhaustion, manifesting as fatigue and dizziness and affecting workplace safety [16]. Hence, with increasing heat waves, accidents at work due to heat stress are growing in Australia [46]. For instance, climate-related heat stress has already been identified as a WHS problem in the construction industry [17,23,58], resulting in accidents on construction sites caused by human factors. Other at-risk industries are agriculture [23,28], forestry, fisheries [28], and the military. However, for many Australian workplaces, heat-related incidents affect not only outdoor workers but also indoor workers in uncooled environments [45,46]. In particular, the increased thermal stress on workers in workplaces without cooling systems in tropical and subtropical regions is a specific characteristic of climate change [11].

#### 3.1.2. Extreme Weather

Studies show that climate change can cause long-term changes in ecosystems and sea-level rise, leading to a possible increase in the frequency and severity of extreme events such as heat waves, droughts, torrential rain, floods, cyclones, landslides, and plant disease outbreaks [4,20,24]. Natural hazards such as wildfires, storms, and floods have always been features of Australia’s unpredictable climate [65]. Tropical cyclones produce destructive winds and torrential rains. Heavy rains can cause flooding, killing over 2300 people in Australia since 1790 [65]. Severe thunderstorms can occur anywhere in Australia, causing hail, tornadoes, and flash floods [65]. Wildfires threaten almost all parts of the country at different times of the year. Intense rainfall triggers landslides [65]. Overall, with climate change, it is expected that communities in Australia and around the world will become increasingly exposed to natural hazards such as storms, floods, and fires, and these changes may evolve into disasters [65].

Climate change will intensify the frequency of extreme weather events in Australia and most other parts of the world [66], with profound implications for human society and the natural environment [4]. Notably, higher average temperatures and longer heat waves will cause catastrophic wildfires and death [44]—in Australia, the 2009 Black Saturday wildfires were the worst on record, claiming 173 lives [65]. Besides injuries and illnesses caused by extreme events such as heat waves, cold spells, floods, and storms in Australia, experienced and anticipated health effects include increased intestinal infections due to the spread of vector-borne diseases [4].

During extreme weather events, workers are at great risk [24], and they assume climate change causes more “extreme weather” and describe the hotter weather as “harsher”, “more extreme”, “aggressive”, and “biting” [44] (p. 266). In particular, hotter temperatures will constantly challenge construction workers and the unions representing and advocating for them [17]. Linked to air pollution, a wave of diseases emerged following Australia’s disastrous 2019–2020 wildfire season and the unprecedented 2022 East Coast floods. Health professionals must be prepared to respond to the public health threats posed by climate change [61]. For example, recent natural disasters such as fires and floods have shown that GPs play a role in disaster recovery [61]. Also, extreme weather events and mega-fires negatively impact mental health [67]. Likewise, images of wildfires, dust storms, floods, or dying livestock shown nationally suggest to many that Australia’s regional areas, such as the outer regions of New South Wales, may not be attractive places to live [64] and work.

#### 3.1.3. Air Pollution

‘Air pollution’ [1] related to climate change was rarely covered in the included literature, and any coverage dealt with the safety and well-being of the general population and not with WHS specifically. Sources mentioning ‘air pollution’ are related mainly to urban areas, where rapid growth and dense construction increase air pollution from vehicles, transportation systems, buildings, and industry. Increased emissions of air pollutants and greenhouse gases affect human health and quality of life [20]. A consequence of increased air pollution is increased lung strain, which can trigger cardiovascular disease, the second leading cause of death during heat waves [21], and chronic health impacts, such as allergies and respiratory diseases [24]. In addition, air pollution, changing rain patterns, and climate extremes can affect people’s safety and well-being [24]. Recent climate events in Australia resulted in health issues that can be attributed to climate change, for example, diseases associated with air pollution following the catastrophic Black Summer Fires of 2019–2020 and the unprecedented East Coast floods of 2022 [61]. Overall, air pollution, together with saline freshwater, wildfires, ozone depletion, water scarcity, and, therefore, reduced crop yields, negatively affects health and the economy and increases migration from unsustainable regions to more fertile areas [66].

#### 3.1.4. Ultraviolet Radiation

Despite increased ‘ultraviolet radiation’ being among the most likely hazards encountered in various environments due to climate change [24], it was scarcely mentioned in the included literature. Studies pointed out the long-term risks of excessive exposure and noted that these are not yet fully understood [49]. In contrast to ‘air pollution’, the studies considering ‘ultraviolet radiation’ (Table 3) did focus on WHS, suggesting further research should focus on impacted outdoor workers who, during the summer, are directly exposed to sunlight. In practice, employers need expert advice on providing information, training, and protective clothing to protect workers from the adverse impacts of ultraviolet light and radiant heat [56].

#### 3.1.5. Vector-Borne Disease and Other Biological Hazards

In Australia, many studies on vector-borne diseases have been conducted [4], some of which were eligible for inclusion in our review [20,24,27]. The results of these studies (Table 3) generally suggest climate change stimulates the transmission of vector-borne diseases, which are transmitted by living agents, such as mosquitoes, carrying the pathogen within the vectors [4]. Since the early 20th century, climate change has increased temperatures, stimulating climate-related diseases that impact health [20]. In particular, climate change affects the life cycle stages of pathogens within vectors, affecting their incubation periods and transmission to humans [27]. Temperature rises have been shown to increase the development of mosquitoes, which shortens the incubation time of pathogens transmitted via this vector; however, it should also be noted that even higher temperatures can lead to a decline in mosquito numbers [4]. Elevated temperatures can cause pathogens to multiply at an accelerated rate, leading to more intestinal infections from food poisoning [4]. Diseases such as the Ross River virus infection, the Barmah Forest virus infection, and salmonellosis adversely affect the health of populations, particularly elderly people [4].

Regarding WHS, workers are affected by vector-borne diseases and other biological hazards in areas with extreme climate events, especially when drainage is poor and increased vector breeding occurs [24]. This is of particular concern in certain geographic areas where vector-borne diseases, such as yellow fever, malaria, dengue, and chikungunya, are sensitive to climatic change and are spreading, affecting large working-class populations [24]. In addition, a significant proportion of extant studies on vector-borne disease and other biological hazards reported results related to Mesoamerican Nephropathy, a disease affecting the kidney that occurs primarily in young and middle-aged male sugar cane workers without traditional risk factors for chronic kidney disease [27].

Because of the positive correlation between higher air temperatures and vector habitat spread, the burden on outdoor workers, particularly in tropical areas associated with vector-borne diseases, will tend to increase sharply as temperatures rise [24]. Evidence relating to the two most studied sectors in terms of productivity indicates that a day with temperatures above 32 °C can reduce daily labour supply by up to 14% due to vector-borne diseases [27]. Overall, based on the current and future spread of vector habitats, it can be concluded that outdoor workers are at increased risk of developing vector-borne infectious diseases [27]. However, the increased risk of vector-borne contagious diseases for outdoor workers has not yet been extensively investigated, and evidence regarding the impacts of climate change on vector-borne infectious diseases in the working population is currently very sparse. A recent study has, however, shown the link between climate change and the increasing frequency of extreme events that move whole microbial communities (bacteria and fungi), including extremely resilient and survivable organisms that can move to different environments far from their origins [27].

#### 3.1.6. Industrial Transitions and Emerging Industries

Climate change has significant implications for industry structure and employment in many locations. Climate change policies have already significantly impacted demand for coal-fired power generation, and the impacts of the closure of such power generation on regional economies and labour markets have been well documented [68]. Further restructuring may arise from changing climate patterns and the capacity to undertake particular forms of agriculture within regions. A decline in one industry within a region may have multiplier effects that impact service industries such as health and welfare, education, and retail trade [68]. Although ‘industrial transitions and emerging industries’ are among the likely hazards that can occur in various environments in response to climate change [24], this risk was not explored in significant detail in the included articles.

#### 3.1.7. Changes in the Built Environment

‘Changes in the built environment’ were scarcely mentioned in the included articles. The term ‘built environment’ refers to buildings, streets, open urban spaces, and infrastructure as the physical components of our living and working environment. The surface microclimate changes with urbanisation, altering radiative, thermal, moisture, and aerodynamic processes [21]. The built environment affects heat stress, which produces physical responses in people exposed to heat for work or exercise. In this way, the built environment can cause health risks, and climate change is likely to increase these risks [21].

Changes in the built environment are among the hazards likely to occur due to climate change [24]. It is primarily heat extremes that challenge the sustainability of built environments, such as houses and workplaces. Heat waves may be more intense in urban areas with a high population density than in the countryside [20]. A changing urban surface affects people and how they live and work. A consequence is the ‘urban heat island’, which is intensified with increasing city size and population density and makes cities warmer on average than adjacent rural landscapes. The ‘urban heat island’ is particularly noticeable at night when it releases the stored daytime heat [21].

The built environment is also an important element of heat stress resistance [48]. Broader education about the built environment, adaptations, and retrofitting against the effects of heat waves can enhance the built environment and the adaptability of the people who live and work in it in the face of climate change. Currently, younger people, in particular, are concerned about a lack of knowledge about the heat-resistant properties of the built environment (34.9%), specific adaptations (up to 50%), and retrofitting techniques (19%) [48]. Planted ‘green roofs’, for example, do little to reduce temperatures but have major potential as they improve the built environment’s quality, reduce dust-related health problems, and lessen noise pollution [62]. An exploratory survey [48] has been conducted, investigating the relationships between heat stress resilience of the built environment and workplace adaptation to assist in evaluating policy changes, government subsidies, and community education campaigns improving heat stress resistance.

#### 3.1.8. Cascading Climate-Change-Related Disasters

Greenhouse gases prevent the Earth from cooling, and if accumulating excessively, they can contribute to climate change by leading to higher temperatures and rising sea levels, which then cause ‘wild weather patterns’, including floods, extreme storms, droughts, and wildfires [66]. A moderate number of the included articles, for example [56], mentioned the ‘compounding’ nature of climate-related hazards, which impact WHS. NSW, Australia, has been particularly affected, given that drought conditions have been followed by the Black Summer wildfires in 2019–2020, the COVID-19 pandemic in 2020–2021, and several flood events in 2021–2022 [69]. On an individual level, some workers were impacted simultaneously by multiple climate-related hazards; for example, during hot days, the pandemic required school outdoor play equipment to be regularly cleaned in full personal protective equipment (PPE) [27]. Explaining the concern with this requirement, a worker in the electricity sector reported that using new forms of PPE to manage COVID-19 risks worsened the effects of heat, leading to increases in skin rashes, dehydration, stress, headaches, and fatigue [50]. Likewise, fireproof workwear to prevent skin cancer multiplied the risks of heat effects:

It is now mandatory in my industry to wear long pants and long-sleeved shirts, and we currently wear flame retardant longs, all of which don’t let the body breathe and cool sufficiently. This is because companies don’t want to be responsible for giving us skin cancer, but I’d prefer to wear shorts and die of cancer in 30 years’ time instead of heat exhaustion tomorrow [50] (p. 71). On an industry level, a human resources director in the timber industry mentioned that single events such as wildfires, COVID-19, and climate change did not necessarily create new bottlenecks. However, the combination of these factors created problems for their business strategy, for example, when wildfires impacted timber exports, and COVID-19 put pressure on global markets [64]. Even in academia, after the Black Summer 2019–2020 and management problems of COVID-19, research outputs and study results were more limited than expected due to the poorer mental and emotional well-being of participating individuals and organisations [67].

### 3.2. Climate Change Impacts on Occupations and Industries in Regional Australia

Research on the impact of climate change on workers and workplaces in Australia has been relatively limited and has primarily focused on heat stress and its impact on labour-intensive and outdoor industries [19]. To date, there has been less focus on other climate change impacts such as air quality, floods, and fire damage and limited focus on industries and occupations not classified as labour-intensive or outdoor work. Furthermore, there has been minimal recognition that the structure of regional labour markets means that regions are particularly vulnerable to the impacts of climate change.

#### 3.2.1. Heat Stress as a Result of Climate Change

Research to date has tended to focus on the impacts of heat stress on the physical well-being of workers, drawing on both survey data [19,59] and workers’ compensation statistics [16,58] to identify days of work lost due to heat stress, correlations between heat waves and injury rates, and other factors that impact workplace productivity. McInnes and colleagues [16] acknowledged that factors outside the workplace may affect workers’ capacities and that heat stress can be cumulative. They explored the impact of high overnight temperatures on the level of workplace injury claims and suggested that poor sleep patterns due to overnight high temperatures may exacerbate accident rates due to heat exposure.

#### 3.2.2. Impact on Specific Industries and Occupations

Among the 41 articles identified for this review, 28 focused on the impacts of excessive heat on outdoor workers, with 23 focusing particularly on the agriculture, forestry, and fishing industries, while 21 examined the construction industry. While researchers have explored the impacts of climate change on specific industries or occupations, Varghese and colleagues [18] emphasised that industry and occupational classifications are problematic as they fail to capture the nuances of the nature or intensity of the work and the locations of regular workplaces. For example, while many workers in the construction industry are outdoor workers and their labour is relatively intense, a significant proportion of workers in the construction industry, such as administrative staff or architects, work indoors and have relatively low-intensity work patterns. In turn, indoor workers in manufacturing, accommodation, and food services may work in conditions where ovens or furnaces are in use, creating a higher heat exposure than might be expected indoors.

A simple indoor/outdoor dichotomy also fails to capture the experiences of those required to drive between work sites or locations. As a result, McInnes and colleagues [16] have developed a more detailed classification system based on categories of work intensity or load and workplace location to capture the nuances of work location. Their location categories include:(i)working only in a vehicle or cab;(ii)outdoors with no work in a regulated indoor climate;(iii)regulated indoor climate;(iv)unregulated indoor climate and vehicle or cab; and(v)indoor and outdoor work.

These categories help to provide a more detailed categorisation of those at risk of heat stress as a result of climate change and may be incorporated into a broader analysis of the regional differences in climate change impact.

#### 3.2.3. Impacts on Regional Labour Markets and Economies

While industry restructuring, labour market transitions, and economic change are acknowledged as potential outcomes of climate change in regional economies [24,31], there has been surprisingly little analysis of the differential impacts of climate change on workers and workplaces in regional and remote locations compared to metropolitan, state and national-level data.

Regions tend to have significantly different labour market structures than urban locations [31]. Industry structure in a region may depend on the region’s natural resources and comparative advantages, and hence, the region may have a narrower range of industries and a flatter occupational structure than metropolitan locations. The age, gender, and skill structure of regional labour markets are also significantly different from those of metropolitan locations [32]. Drawing on ABS Census data [32], categorised into metropolitan, regional, and remote geographical regions, we find that employment in remote locations is dominated by agriculture, forestry and fishing, mining, healthcare, and construction (The Australian Statistical Geography Standard (ASGS) Remoteness Structure defines Remoteness Areas for the purpose of releasing and analysing statistics. Remoteness Areas are derived from the Accessibility/Remoteness Index of Australia Plus (ARIA+), produced by the University of Adelaide. Remoteness Areas divide Australia into five classes of remoteness on the basis of a measure of relative access to services. The five remoteness classes are Major Cities, Inner Regional, Outer Regional, Remote, and Very Remote [70]. In this study, Inner and Outer Regional, and Remote and Very Remote have been combined to create a three-category classification system, Metropolitan, Regional, and Remote, that captures key elements of analysis). There are significantly lower employment levels in financial and insurance services, media and communications, and retail trade.

This contrasts with metropolitan locations where agriculture and mining employment levels are low, and the highest employment levels can be found in healthcare and social assistance. Metropolitan locations also have a broader range of industries, such as retail trade, professional, scientific, technical services, education, and construction, all of which provide similar employment levels.

In regional locations, there are higher than national average levels of employment in agriculture, forestry, fishing, and mining and lower than average levels of employment in finance, professional, scientific, technical services, and media and communication. The Census data [32] also show that workers in remote locations are more likely to be employed as labourers, machinery operators, tradespersons, and technical workers than the national average and less likely to be professionals and sales workers than metropolitan workers. The high level of managers in remote locations reflects the high proportion of workers who are farm owners and managers.

Remote and regional workers are more likely than metropolitan workers to be employed in industries with a high level of outdoor work, which are considered more labour-intensive. The occupational structure of remote locations also reinforces this tendency, with machinery operators more likely to work in vehicles, cabs, or outdoor locations. In regional and remote locations, even industries that are generally considered indoor work, such as healthcare and social assistance, include a higher component of community outreach functions, which may require workers to spend extended time in vehicles, cabs, and unregulated indoor locations (such as client homes) [4].

Bi and Parton [4] note that regional and remote locations are also less likely to have the same range of resources for cooling opportunities as metropolitan locations. Remote locations are less likely to have large air-conditioned community facilities, such as shopping malls or libraries, and workers may have to travel significant distances to access municipal pools or other facilities.

Hyatt and colleagues [11] have documented significant regional variation in changing heat patterns. Similarly, Hall and colleagues [8] have shown that some regional and remote locations in Australia may be significantly more impacted by changes in heat conditions due to climate change. Also, regional and remote locations may be increasingly susceptible to a broader range of climate change disasters than metropolitan locations, with increasing incidences of wildfires, floods, and cascading disasters [30].

### 3.3. Climate Change Impacts on the Health of Australian Workers

When considering climate-related health impacts on Australian workers, the included literature focuses on heat impacts, while further climate hazards are largely neglected. Hence, the narrative in the sample literature around Australian workers’ health impacted by climate change is predominantly created around ‘raised temperatures’.

#### 3.3.1. Physical Health

The risks to physical health associated with climate change are not particularly obvious, at least to the lay public, even though there is evidence that both cold and hot temperature extremes are associated with high mortality rates [71]. It has been identified within Australian occupational settings by occupational hygienists and other specialists that workplace heat exposure presents a growing threat to health and safety, particularly with respect to preparedness for heat waves and prevention measures [55].

A particular issue that has arisen as a significant threat to health as identified by Australian workers is that regardless of jobs or location, about 90% of workers might be affected by heat on hot days, compared with only about 10%, indicating that they may never or rarely be affected [44]. Although the reasons for this reporting are varied and highly dependent on the physical nature of the working environment and the type of job, the number of workers’ compensation claims has been shown to increase as the severity of the heat waves encroach toward moderate-to-high compared with non-heat wave days, with the vulnerability being skewed towards males, workers under 34 years of age, medium-heavy strength jobs, and the industrial sector [18]. In addition, there appears to be an increase in injuries ranging from musculoskeletal to burns, lacerations, amputations, and internal organ damage associated with moderate to extreme hot temperatures [12]. An interesting observation by these authors is that claims related to heat illness seem to be associated with an ambient temperature of 35.5 °C or more, where every additional 1 °C increase is then associated with a 12.7% increase in claim incidence. Overall, heat-related severe illnesses and injuries are not uncommon in Australian workplaces [20,59].

The increase in morbidity and mortality with heat extremes is broadly attributed to the general response to higher temperatures and their physiological implications, considering the particular climate and population characteristics [9]. Broadly, the mechanisms of injury during heat extremes have been identified as fatigue, reduced psychomotor performance, loss of concentration, and reduced alertness [28]. However, Ebi et al. [21] underscore that the primary cause of death during heat waves is cardiovascular illness, with kidney dysfunction as a consequence of dehydration potentially worsening cardiovascular symptoms, and respiratory illness being the second greatest source of mortality and morbidity during heat waves.

A critical finding that has emerged in the literature is whether the threat from heat waves is relative to indoor or outdoor occupational settings. Intuitively, outdoor workers are more likely to be exposed to extreme heat during Australia’s summer months, resulting in widespread adverse effects on health and productivity [23]. Although the data from this study suggested that heat exhaustion was a common occurrence, it was not perceived as important by management and workers compared to the loss of productivity due to the slowing of pace. This observation has been challenged as problematic since the slowing of pace or loss of productivity is most commonly attributed to the effect of external heat load. However, it has been suggested that bodies not only absorb heat but also produce heat, resulting in a co-production of heat [54]. These authors found that manual workers in the Australian monsoon tropics were able to reduce the heat load by adapting and modulating the elements of their work practices, which enabled work to be conducted without incurring additional heat-related illness. This outcome indicates that the risk of injury during extreme heat events can also be curtailed or at least mitigated to some extent with adaptive work strategies.

In contrast, data relating to the exposure–response relationship between ambient temperatures and occupational injury in city workers located in Greater Adelaide, South Australia, indicated that younger (<45 years) male workers required to perform heavy physical work, predominantly outdoors, were at higher risk of occupational injury during moderate and extreme heat [16,47]. Although the effects of additional heat load on morbidity and mortality among workers are well described [24], there are risks beyond heat illness and injury that have been found to be of concern. For example, an increase in ambient temperature has been observed to have a positive effect on the development of mosquitoes, in addition to shortening the incubation period of the pathogens within vectors and increasing the duplication of enteric infections [4,27]. Finally, Ebi et al. [21] found that the physiological factors associated with increased risk of death based on epidemiological observations with evidence-based explanations include cardiovascular, respiratory and cerebrovascular factors, as well as genitourinary diseases, diabetes, and hypertension. In addition, these authors found that fatal exertional heat stroke has occurred in various sporting contexts. Overall, life-threatening cases of exertional heat stroke are ten times more common during warm-weather endurance events.

#### 3.3.2. Mental Health

Although the relationship between climate change and increases in mortality and morbidity is well established, the effects on mental health capture less public attention, even though there is good evidence that exposure to high heat stress worsens mental health, with a list of symptoms including irritability, dizziness, confusion, fainting, and seizures [7]. It has also been found that at temperatures above 32.2 °C, there is likely to be a significant drop in mental performance, including speed of response, reasoning ability, visual perception, associative learning, and mental alertness, which has been suggested to be one of the potential causes of fatal accidents [51]. Further to this, a cross-sectional survey of perceived heat stress symptoms on work-related tasks in Australia’s Monsoon North found heat to exacerbate pre-existing mental health issues, suicide rates, and violence, while 25% of workers reported a negative impact of heat on relationships with friends and family on a daily, weekly, or fortnightly basis [60]. Most disturbing, however, was that this study concluded that the effects of mental health during times of peak temperatures and humidity were associated with the prevalence of homicide, serious assault, and suicide rates, along with increased hospital admissions for behavioural and mental health issues.

In a survey of 547 workers across both indoor and outdoor occupations, typical physical injuries were reported. Notably, the workers also indicated that irritability due to heat led to arguments among co-workers [50]. Beyond the potential adverse interaction that excessive heat might cause between co-workers, it has also been found that the environmental workforce is at particular risk of mental distress beyond the usual job stressors since these individuals are exposed to traumatic events (e.g., wildfires) and trends (e.g., deforestation), ecological grief, distress, and overcommitment to work [67]. Similarly, extreme weather events and related community displacement can cause mental health distress [61]. Finally, an issue that has recently been shown to have serious long-term mental health consequences due to excessive heat is the loss or disruption of sleep [60]. Carter et al. [60] found that diminished sleep was reported in 29% of workers, resulting in tiredness and declines in cognition, which can be hazardous in settings where heavy machinery and vehicles are used as part of the job.

### 3.4. Socio-Economic Impacts of Climate Change

Climate change, particularly higher temperatures causing natural disasters, extreme heat, and heat stress, has severe socio-economic impacts. The literature defines socio-economic impacts generally as “the effects of development as perceived and experienced by local residents” [72] (p. 61). These ‘effects of development’ are based on social and economic factors relevant to residents and, more broadly, to an economy or even the global economy (e.g., global gross domestic product). Such socio-economic factors are numerous, including gender, age, income, wealth, education, occupation, and health [73,74].

In the context of this review, four main socio-economic themes emerged: health issues, impact on performance and productivity, impact on the economy and society, and workplace issues. The literature predominantly discusses these themes as adverse effects of climate change developments. However, a few articles also propose mitigation actions while showing potential for further exploration and suggestions. Notably, most of the included literature discussed climate change effects largely in the context of heat.

#### 3.4.1. Health Issues of Workers

Workers exposed to (extreme) heat are likely to suffer from numerous health issues. Fatigue, illness, work stress, physical discomfort, morbidity, accidents, and mortality—temperatures exceeding 39 °C can kill [20]—are the most frequently highlighted physical problems [20,24,47]. Also, cognitive impairment, reduced psychomotor performance, loss of concentration, reduced alertness, and heat-related disorders adversely impact workers’ health in industries with high heat exposure [28,60]. In Australia, such industries include agriculture, construction, mining, FIFO (fly-in/fly-out) occupations, and the military [19,62]. Hence, climate change and hot weather are major growing challenges to workers’ health and safety in these high-intensity outdoor industries.

In mitigating these adverse health risks and issues, the literature suggests, wherever possible, shifting work to cooler times of the day, applying cooling systems, and offering cool locations for rest breaks [9,19]. In addition, the frequency and duration of rest breaks should be adequately organised, allowing self-pacing to enable workers’ bodies to cool down and recover to an optimum functional and work capacity [9,10,46]. Also, supplying plenty of fresh water for adequate hydration is a critical protective measure. However, at some workplaces, a lack of access to toilet facilities (e.g., road construction) influences the workers’ hydration status [46]. For example, in an interview in Hansen et al.’s [46] (p. 7) study, a participant said they did not eat or drink when at work due to no toilets being available, and hence, “all the time at work, I’m dehydrated, constantly”.

As a result, while some mitigation action exists, more specific and additional solutions are needed. Workers’ negative health implications of climate change impact job satisfaction, lower morale, and lead to higher staff turnover. Additionally, illness, injuries, accidents, and even deaths increase costs to industry (e.g., legal and insurance). Consequently, organisational productivity decreases. Overall, the narrative that emerges from the literature is that there is a tension between the need to protect workers’ health and organisational priorities to maintain productivity in the work environment [11].

#### 3.4.2. Impact on Performance and Productivity

The included literature revealed that reduced performance and productivity are major consequences of climate change, principally due to increased heat stress affecting workers and resulting in health risks and financial issues. Reduced performance and productivity start at the worker level and cascade across the organisation.

Generally, increasing temperatures are related to decreasing labour productivity [7,62]. Extreme heat during work produces occupational health risks and limits a worker’s physical functions, capabilities, work capacity, efficiency, and productivity [20,26,27]. For example, workers operating at 33–34 °C and with moderate work intensity can lose about 50% of their work capacity [20]. Consistent with this, a shearing contractor interviewee stated that “the average shed output during comfortable weather would be 600 sheep a day. On the hottest days in summer, this drops to 400” [23] (p. 244). Hence, high temperatures during work impact performance, requiring measures such as self-pacing and taking extra and more frequent rests, resulting in reduced worker activity and labour productivity. For example, 70% of Australian workers reported reduced work efficiency from heat stress on at least one day a year, costing USD 932 annually per person (USD 656 per person across the entire workforce) [26]. In tropical areas of Australia, in particular, the effects of heat are a significant issue for employees’ health and productivity [27].

Heat affects the total productivity of the labour force through reduced labour supply, labour effort, and labour productivity, and heat-related productivity loss is one of the most prominent ‘market impacts’ [62]. In Australia, for example, workers do about one hour less work daily when temperatures exceed 37 °C, compared with days when the temperature is below 30 °C [27]. Unfortunately, heat stress is also frequently accompanied by other challenges, such as a lack of social protection and working poverty.

Policies (e.g., workplace heat management policies), guidelines, and heat management systems could alleviate such challenges and productivity loss by protecting the health of working people and their individual productivity, including their income [11,26]. Other strategies to mitigate productivity loss include, for example, shifting work schedules two hours earlier (e.g., from 8 a.m.–5 p.m. to 6 a.m.–3 p.m.) to reduce costs by about 33% [75]. Similarly, changing business hours to cooler seasons or within a given day appears efficient. However, such an approach is associated with reduced overall working hours and related income, thus affecting lower-income workers the most [62]. Furthermore, studies suggest that providing shade for workers decreases costs by factors of six and ten, respectively [26].

In conclusion, decreased labour productivity leads to less economic production, impacting regional, national, and potentially global prosperity.

#### 3.4.3. Impact on the Economy and Society

Estimates of the impacts of climate change on work capacity indicate that heat-related productivity losses will occur in many regions of the world [11] and, hence, also impact society and economies in Australia. Where possible, an alleviating approach could include shifting economic activity from sectors most exposed to high temperatures to sectors with low exposure; for example, reducing farm employment and increasing industrial or service jobs, impacting workers and economic outputs, including the aggregate productivity of modern, diversified economies [62].

In contrast, climate change with rising temperatures may increase labour productivity in regions with low baseline temperatures [62]. Hence, shifting economic activities may occur cross-regionally and even cross-nationally.

Although economic transformations are rarely driven by climate change [62], climate change adaptation and mitigation strategies must be implemented to minimise future costs [26]. For example, a substantial cost could be avoided by offsetting occupational heat stress costs [76] and if global warming is limited to less than 1.5 °C [77].

Climate-related natural disasters like drought, wildfires, and flooding cause further economic damage. For example, Australian society, especially in communities in rural regions, is becoming more vulnerable to natural disasters, at least in relation to economic costs [4]. Any intensification or frequency of events, such as floods or severe storms, will likely significantly affect disaster costs. However, adaptive planning can minimise damages based on systematically analysing climate change’s economic and social costs in certain Australian regions [4].

Besides economic influences, climate change and, consequently, extreme heat also impact society, as such developments regularly result in reduced personal and family income and nutrition, negatively influencing the quality of life [9,47] and raising questions of social justice [20]. For example, in an Australian study, an interviewee from the construction industry commented, “No work, no pay” [46] (p. 8). Even though self-paced, frequent rest breaks are acknowledged approaches to heat protection, practice proves different. There may be underlying economic incentives where contractors or self-employed workers only get paid when the job is satisfactorily completed, meaning that “[t]hey’ll just work right through, no matter what the conditions, and do it as quick as they can” [46] (p. 8). Consequently, such practice can lead to unsafe behaviour and accidents [28].

To mitigate such situations and thus protect working people’s health and contribute to society’s prosperity, policymakers can play an important role by setting and enforcing guidelines and codes of conduct.

#### 3.4.4. Workplace Issues

Policies must be translated and embedded in an organisation’s processes to be effective in the workplace. For example, workplace heat management policies contribute to better worker health and safety and reduce future heat-related economic burdens [26]. Under the WHS umbrella, some Australian sectors (e.g., mining, construction, and military) use heat stress risk assessments and prevention plans to alleviate reduced worker productivity and related higher workplace accident rates [19].

However, protecting workers from hazardous heat exposure appears challenging for many Australian workplaces. Productivity demands and higher costs, such as heat-related healthcare expenses or energy costs (e.g., for cooling systems), necessitate balancing these with WHS imperatives [50]. Yet workers’ health, safety, and welfare in extreme heat must not be jeopardised [45]. Acclimatisation is an approach Australia uses to balance WHS in hot places better, selecting workers who are likely to be heat-adaptable in a specific occupation [19]. Potentially higher recruitment costs are likely to be compensated with, for example, higher labour productivity, fewer accidents, better retention rates, and more contented workers [19].

On the other hand, recruiting poorly acclimatised workers who lack experience working in the heat negatively affects workers and businesses. Zander et al. [19] (p. 10), therefore, ask the question, “Why, in some hot and humid parts of Australia, more people from a similar climate are not recruited, but instead, a large part of the labour force are FIFO workers from cooler parts of the country”. While qualified worker shortages and a low unemployment rate in Australia may partly answer this question, further investigation will be needed.

## 4. Discussion

This scoping review was designed to provide an overview of what is known about climate change impacts on WHS in Australia and to identify key gaps in knowledge of this topic area. The findings highlight the broad range of impacts climate change is having and is predicted to have on WHS in the Australian context. The findings also reveal that climate change impacts on WHS can, in turn, affect worker and sector productivity and industry and society more broadly, influencing the economy, labour markets, and living conditions. Nevertheless, despite the importance of these issues, it is also evident they have received limited attention in research focused on Australia. Notably, there are numerous types of climate change impacts and industries in Australia that have been subject to little or no research in relation to WHS. Increases in ambient temperatures and extreme weather were the two impacts of climate change most frequently identified in the included literature. This may be due to the fact that more people die from extreme heat than from all other natural hazards combined [78], and according to climate change projections, temperatures and heat-related mortality will continue to rise [4,15,79].

Other globally recognised impacts of climate change, including increases in air pollution, ultraviolet radiation, vector-borne disease, industrial transitions, emerging industries, changes in the built environment, and cascading disasters, were considered in only a small number of the included studies, with a focus on Australia. Similarly, outdoor workers, particularly those from the agriculture, forestry, fishing, and construction industries, were most heavily represented in the included literature, with indoor workers often considered a comparator group. However, climate change is likely to be impacting numerous other industry sectors of importance in Australia, like arts and recreation, education and training, healthcare and social assistance, wholesale and retail trade, and public administration and safety, which have been researched far less frequently or not at all. Finally, our review identified greater knowledge of climate-related diseases than injuries, a finding consistent with conclusions drawn by Varghese et al. [28]. These identified gaps in knowledge highlight an urgent need for further focused research to inform strategies that will enable Australian industries to adapt to climate change.

The review found that the main focus of previous research on climate change impacts on WHS in Australia has been heat and raised temperatures. Excessive heat loads in occupational settings can cause illnesses, particularly if strategies to reduce heat stress are inadequate. For example, inadequate fluid intake can increase heat stress and impair physical performance [54]. Fatal chronic kidney disease was found in sugar cane harvesters in Central America, triggered by daily dehydration due to excessive sweating in the hot work environment [9]. Generally, extreme heat increases death rates: the two weeks of extreme heat in France in August 2003 caused over 1000 additional deaths among people aged 20–70 years [9]. Some regions, such as Southern Asia and Western Africa, are expected to be the worst affected by the adverse consequences of global warming. With global warming of 1.5 °C during this century, extreme heat is predicted to cause the loss of about 9 million full-time jobs by 2030 (about 4.8% of all working hours) [20]. Overall, the health, social, and economic losses during unusually intense heatwaves could be substantial in Europe and North America [20]. It is likely that Australian industries, particularly those in hotter regions of the country, will be similarly impacted, given the heat concerns identified in this review.

The review reveals a further focus of previous research on the impacts of climate change on WHS in the Australian context has been on outdoor workers such as those working on construction sites or in agriculture, though some knowledge has also been gained regarding adverse impacts due to climate change on indoor workers, particularly in workplaces with unregulated temperatures—with the majority of this research relating to heat stress. The findings, for the Australian context, are consistent with those globally, which indicate heat stress is a serious issue for many of the world’s 66 million textile workers and 1 billion agricultural workers and for other workers employed, for example, in emergency repair work, transport, refuse collection, sports, and tourism [20]. In a similar vein, most of the working hours lost to heat stress in Europe, North America, and the Arab States are lost in the construction industry [20]. The military sector has also been considered, along with these other industries exposed to high temperatures. However, we found that specific research on climate change impacts relating to heat stress in other occupational settings is scarce in Australia, which parallels the global findings of Kjellsrom et al. [20].

As climate change progresses, occupational health problems are likely to increase further in high-income countries, such as the United States [9] and Australia. A previous scoping review on climate change and mental health revealed that climate-related exposures to hazards, such as drought, wildfires, and floods, were linked to psychological distress, reduced mental health, and increased suicide rates [80]. Overall, our review found that climate change is contributing to more occupational diseases than just injuries alone in Australia. Taken together, this knowledge from Australia and other countries makes it remarkable that, globally, only 34% of people of working age have insurance in the event of an occupational injury [20]. A study by Varghese et al. [12] reported that in South Australia, about 5% of all compensation claims were attributed to temperature, an estimate that is well (2.7%) above similar estimates from Spain [71].

While in Australia, workers currently do about one hour less work per day when temperatures exceed 37 °C, as compared to being below 30 °C [27], it is projected that by 2030, the equivalent of more than 2% of the total working hours worldwide will be lost annually, either because it is too hot to work or because workers have to work at a slower pace [20]. Moreover, in poorer countries, such as Southern Asia and Western Africa, productivity loss may even reach 5% [20]. Unfortunately, heat stress is often accompanied by other challenges, such as a lack of social protection and working poverty, which are more prevalent in countries with work deficits.

Estimates of the impacts of climate change on work capacity indicate that heat-related productivity losses would occur in many regions of the world [11], impacting society and economies in Australia and across the globe. For example, a global temperature increase of 1.5 °C by the end of the 21st century and labour force trends lead to projections of a productivity loss of 80 million full-time jobs by 2030 [20]. The global economic burden will be substantial, with costs to the economy and society likely to be severe in poorer or developing regions, as they generally have higher baseline temperatures [62].

While climate change impacts rising temperatures, causing significant issues with productivity loss in certain regions, as discussed above, it should be noted that it may also increase labour productivity in regions with low baseline temperatures [62]. Hence, shifting economic activities may occur cross-regionally within Australia and other countries and even cross-nationally, fostering global collaboration. The trend of globalisation supports such approaches, despite emerging shifts in opinion toward deglobalisation since the onset of the COVID-19 pandemic [81] and recent geopolitical events.

Kjellstrom et al. [9] investigated heat exhaustion and reduced human performance and stated that later in the 21st century, many of the four billion people living in hot areas worldwide will experience considerably reduced work capacity due to climate change. They further claim that 30–40% of annual daylight hours will become too hot for outside work in some areas. Consistent with this, the findings of our review indicate some traditional work hours may similarly become untenable for many occupations in Australia, as currently performed, due to heat stress. As a result of the impacts of climate change in Australia and across the world, the social and economic impacts will be substantial, with global gross domestic product (GDP) losses predicted to be greater than 20% by 2100 [9].

The estimated productivity reduction due to climate change impacts at the population level is 0.2% for Australasia, which is relatively low compared to other highly affected regions, such as South Asia (11.5%). Similarly, the Caribbean faces a reduction of 11.7%, Oceania (15.2%), Central Africa (15.4%), West Africa (15.8%), Southeast Asia (18.2%), and Central America (18.6%) [9]. Even with a shift in workforce distribution from heavy physical work to lighter service jobs, there is likely to be a significant decline in labour productivity in some regions and countries, and Australian industries, like those in other countries, will need to take steps to counter this.

Besides economic influences, climate change impacts society more broadly, as the findings of our review indicated. Globally, excessive heat levels are often more prevalent in countries with work deficits, often associated with a lack of social protection and working poverty, aggravating inequality between rich and poor countries and population groups within the same country [20]. Such emerging situations habitually result in reduced personal and family income and nutrition, negatively influencing the quality of life [9,47] and raising questions relating to social justice [20]. Within Australia, it is well known that levels of socio-economic disadvantage vary widely across regions and population groups, and some of these issues, noted globally, will, therefore, be important to consider and manage in the face of climate change.

As noted, the human factors impacting productivity in Australia and other countries can range from heat-related illnesses to fatigue, leading to impaired physical and mental performance [51]. There is a complex ‘interplay’ of climate change impacts on WHS regarding physical and mental health and socio-economic costs. Overall, strategies will be needed to ensure the health protection of workers (e.g., more and longer rest periods and active cooling strategies) while maintaining productivity in the work environment. That is, without additional strategies, climate change will mean that less productive work hours can be safely delivered, and there will be increased company costs relating to WHS and workers more broadly as organisations seek to sustain productivity (e.g., wages, air-conditioning, insurance, sick days).

This review identified numerous gaps in knowledge regarding the impacts of climate change on WHS in Australia. In particular, Table 3 shows that regional and remote Australia have received little attention in research investigating hazards associated with climate change and that some kinds of hazards have rarely been the focus of research in the Australian context. Complementing that table, Table 4 provides an Australian ‘catalogue’ of topics explored in research regarding climate change impacts on WHS. Based on the findings of our review, the catalogue summarises topics that have been investigated in existing research pertaining to the Australian context and topics that have received little or no attention and warrant further research. 

In Australia, industries recognised to be associated with high heat exposure include agriculture, construction, mining, FIFO (fly-in/fly-out) occupations, and the military [19,62]. These are somewhat similar to European industries acknowledged to be associated with high heat exposure, namely manufacturing, construction, transportation, tourism, and agriculture [27], though with some notable differences. Yet, in Australia, little research on the impacts of climate change has been conducted in industries other than agriculture, forestry, fishing, and construction (Table 4). Furthermore, it is likely workers in many of the other little-researched industries represented in Table 4 are also vulnerable to the WHS impacts of climate change, and this warrants further research.

Globally, awareness is growing that climate change adaptation and mitigation strategies must be implemented to minimise future costs [26]. Examples include mitigating or avoiding occupational heat stress [76] and taking steps to limit the increase in average global temperature to less than 1.5 °C [77]. In rural and regional Australia, communities are becoming more vulnerable to natural disasters caused by climate change; hence, adaptive planning is needed to minimise damage, which requires a systematic analysis of the economic and social costs of climate change.

Policies, guidelines, and heat management systems can also protect workers’ health, productivity, and income [11,26]. Some Australian sectors (e.g., mining, construction, and military) use heat stress risk assessments and prevention plans to support WHS and worker productivity [19].

From a practical point of view, the literature suggests strategies such as shifting work hours to cooler times of the day or seasons, working in the shade, applying cooling systems, supplying plenty of fresh water, and offering cool locations for rest breaks [9,19,26], while allowing work and rest patterns to be self-paced so workers’ bodies can maintain an optimum work capacity [9,10,46].

However, the current literature review also identified that measures like these that support workers’ health come with a cost, particularly for companies. Hence, there is competition between workers’ health protection, investing in mitigation measures, and maintaining productivity in the work environment [11].

This scoping literature review contributes to practice in various ways. In terms of practical contributions, the ‘catalogue’ of topics provided in Table 4 can be used by various WHS stakeholders. Policymakers can use the catalogue as a source of information when developing WHS policies. In this way, the catalogue provides under-researched occupations with some visibility. Organisations may also use it to guide discussions about climate change and WHS with their stakeholders, particularly in geographical areas and occupations at risk. The catalogue illustrates the various elements (hazards, occupations, health, and productivity) that should be considered when responding to the impacts of climate change on WHS, which are interrelated. As such, it may help organisations develop a holistic view that acknowledges the multifaceted nature of climate impacts on WHS and enables them to provide adequate measures and support for their leaders and workers. This, in turn, has the potential to move organisations forward in becoming safe and sustainable workplaces when facing climate change.

One of the major recommendations for practice that emerged from the review is for the creation of clearer and more robust policies to protect workers from serious occupational health and safety issues [17,20,55]. Our review identified strengths and gaps in the literature, which can inform future policy and guideline development [56]. Organisationally imposed policies, however, may lead to diminished self-determination in terms of workers’ motivation to reduce risk in their workplace environments [45]. Therefore, ‘shared responsibility’ in every workplace could include seeking workers’ perspectives on WHS planning and ensuring the guidelines and policies are job-specific and targeted at vulnerable workers [50]. Workplace policies offer the opportunity to create safe workplace settings where workers understand and actively mitigate risks [23].

A further recommendation for practice is for change in social practices; for example, instead of installing additional air conditioners, change the focus from “when/if work is conducted to *how* work is conducted” [54] (p. 891). For instance, this might involve shifting working hours from the hottest parts of the day to cooler hours, particularly for outdoor work, considering that shifting work patterns can also cause health impacts [62]. For example, introducing additional night shifts was found in Qatar to pose further risks to workers getting insufficient sleep [82]. Such structural changes are mainly recommended in the transport, energy, construction, and agriculture sectors [20]. However, simply allowing for more rest is impracticable in many industries [23]. Hence, social dialogue can be crucial in finding appropriate solutions [20]. Overall, structural reforms are suggested, and in some instances, such reforms may include an overall strategy to support worker transition to other sectors and ensure economic development despite climate change hazards [20].

Workplace training is also recommended to enable workers and other stakeholders to better understand concerns like heat-related illnesses and the severity of heat stroke in hot working conditions [45], particularly in the construction and agriculture sectors [20]. Such training is especially commended, as it comprises the primary source of information for workers on climate-related hazards and their management [63]. Health interventions and education may adopt a holistic strategy targeted at both employers and workers, promoting safe health behaviours [23]. For example, an international comparison shows that in the USA, India, and South Africa, most outdoor physical workers understand reasonably well the symptoms and severe consequences of excessive exposure to heat [63].

Finally, preventive measures such as heatwave forecasts are recommended to minimise occupational risks to workers [18], in this example, by flagging a requirement to request they slow down or interrupt their work [54]. In 2010, California became the first US state to enact a heat-specific law protecting workers from heat exposure. However, after two years, many of the employers audited were not complying with the new heat standards (e.g., drinking water frequently, wearing light-coloured and translucent clothing, taking breaks in the shade, and responding to early symptoms) [63].

This scoping literature review also contributes to theory in various ways. The central theoretical contributions of this work consist of the ‘conceptual map’ (Figure 2) and the ‘catalogue’ of topics (Table 4), which summarise the coverage of extant literature in terms of climate hazards, occupations at risk, health impacts, and socio-economic impacts. Such a map and catalogue help identify areas in which little research has been conducted and thus can serve as a starting point for future research. Furthermore, the map indicates the interconnectedness between the main topic areas (hazards, occupations, health, and productivity) and can be used as a guide when further investigating these relationships. Based on this, Schulte et al.’s [2] framework can be further developed by integrating a ‘cascading’ component that recognises these interconnections and the flow-on impacts of climate change. While our review presents another example of research being recommended as a ‘priority for action’ [1], our catalogue (Table 4) goes further by identifying specific areas in need of further research and comprising a tool that can be used to guide WHS risk assessment and risk management in the face of climate change.

Additionally, this review makes some methodological contributions. First, various literature reviews on climate change impacts on WHS have already been conducted [10,24,26,27,28]. The novel aspects of this review include the use of the scoping review methodology, which allowed us to ‘map’ important literature in the area of interest [35]. Second, our review combines a scoping literature review approach [35] with template analysis [52]. We demonstrated how template analysis, predominantly used to explore empirical data, can be employed together with the NVivo V12 software application to analyse scholarly literature. Finally, combining elements from different review approaches, such as using peer-reviewed articles in combination with other reports [39], allowed the integration of the disciplinary silos of ‘climate change’, ‘health’, and ‘WHS’ in Australia. Such integration, in turn, uncovered related areas such as ‘climate-related hazards’, ‘occupations at risk’, ‘WHS outcomes’, and ‘socio-economic factors’, which need to be addressed together in research on climate change impacts on WHS.

Based on the review’s findings, several recommendations can be made for future research on climate change impacts and risk mitigation strategies in WHS in Australia. One study [19] identified acclimatisation as an approach that could be more widely adopted in Australia to manage WHS risks in hot locations by recruiting workers more selectively and ensuring their capacity to be heat-adaptable in a specific occupation. There is potential to obtain more insights into this risk mitigation strategy, including investigating benefits such as well-being at work, job satisfaction and motivation, productivity, and physical discomfort. Furthermore, quantifying and contrasting the benefits against potentially higher recruitment costs would be worthwhile. As regions in Australia vary widely in baseline temperatures, research is also needed on where and how economic activities can be shifted cross-regionally to increase productivity and reduce heat stress concerns for workers as climates continue to change. Overall, future research should focus on climate hazards other than heat and prioritise research in new areas, such as cascading weather events and air pollution associated with climate change and their impacts on WHS.

Similarly, occupations such as agriculture and construction have received research attention. However, there is scope to investigate areas with less research coverage, such as uncooled indoor workplaces such as restaurant kitchens and workshops. Our review highlighted how the impacts of climate change on WHS, including climate-related hazards, occupations at risk, and the health and productivity of employees, are interrelated. Further research on this ‘interplay’ could provide a better understanding of the multi-layered phenomenon of climate change impacts on WHS. Further research is also needed to explore the differential impact of climate change on regional and remote locations, broadening the focus from metropolitan and national-level data, which are currently most common. Finally, in taking a meta-perspective, a future research avenue may include investigating fundamental potential reasons for the gap in the literature, such as that researchers do not regard the topic as important, funding agencies do not think the topic is relevant, and the topic is not understood to be feasible to study.

The review has three main limitations to be acknowledged. First, our review focuses on extant literature pertaining to the Australian context, which is relatively sparse. Much further research is needed, but to address this limitation, we have considered knowledge and frameworks in the global literature when developing the ‘catalogue’ of topics in Table 4 and discussing possible options to address key issues. Second, the results of our analysis focused on selected thematic clusters, chosen by the frequency of occurrence of the emerging themes within the available and included sources. Additionally, less prominent clusters present in the Final Template (Appendix A), such as ‘Prevention and Preparation’, could be considered in future studies but were beyond the scope of this review. Finally, our dataset was derived from 41 journal articles and other scholarly works identified by using four search terms (climate change; workplace; health; Australia) and five scholarly databases (CINAHL Plus/EBSCO Host; Emcare (Ovid); MEDLINE ALL; Health Collection (Informit); and PubMed). There is potential to extend both the databases and search terms employed in future reviews.

## 5. Conclusions

This scoping review explored the extant literature on climate change impacts on WHS in Australia. It mapped the coverage of climate hazards, occupations at risk, and health and socio-economic impacts within the available literature and identified key gaps in knowledge. The main themes derived from the included studies were heat and extreme weather events as key climate hazards; agriculture and construction as some of the better-researched occupations at most risk; physical and mental illness and injuries as important adverse WHS outcomes; and loss of productivity as an important industry outcome. The analysis revealed that these main themes are interconnected, which adds complexity and importance when considering the impacts of climate change on WHS within Australia. Hence, it is essential to look at WHS holistically, considering all of these main elements (hazards, occupations, health, and productivity). Such a multifaceted view of WHS will help workers, organisations, and policymakers understand the risks, strategies, and supports needed to enable adaptation and resilience in the Australian industry in the face of climate change. The findings of the review and the mapping it provides can be used to start discussions on overlooked regional sectors and serve as a starting point for empirical research in under-investigated areas, supporting these industries to identify and respond to climate change impacts.

## Figures and Tables

**Figure 1 ijerph-20-07004-f001:**
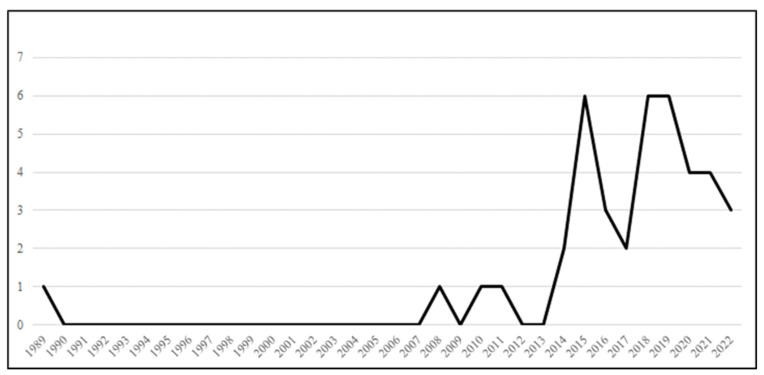
**Publications in the field of climate change impacts on WHS in Australia (1989–2022)**. Eligible articles per publication year (N = 41).

**Figure 2 ijerph-20-07004-f002:**
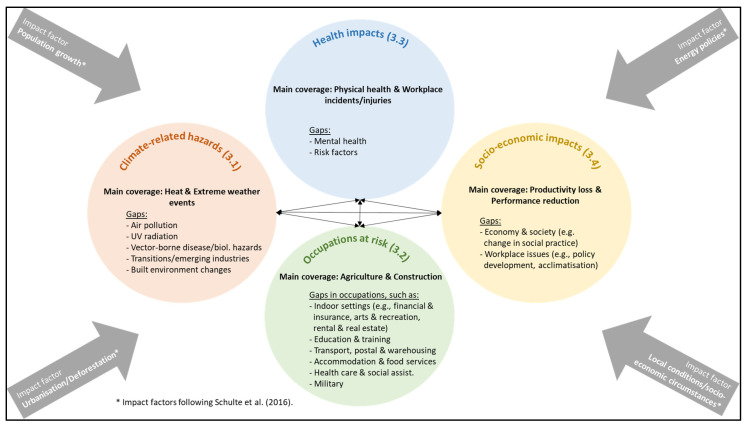
Conceptual map illustrating research coverage of climate change impacts on WHS in Australia and associated knowledge gaps. * Arrows indicate impact factors discussed by Schulte et al. (2016) [1], which influence interconnected climate-related areas of WHS in Australia (circles).

**Table 1 ijerph-20-07004-t001:** Search strategy: databases, terms, and outcomes.

Database	Phenomenon: Climate Change	Concepts: Workplace, Health, Australia
Cinahl Plus/EBSCO Host	“climate change”+	workplace + health + Australia (11) -> included 7
Emcare (Ovid)	“climate change”+	workplace + health + Australia (23) -> included 8
MEDLINE ALL	“climate change”+	workplace + health + Australia (1) -> included 0
Health Collection (Informit)	“climate change”+	workplace + health + Australia (19) -> included 9
PubMed	“climate change”+	workplace + health + Australia (27) -> included 7
Expert advice	Sources suggested by experts in the areas of environmental economics and healthcare	9 -> included 4
Reference lists	Sources cited in leading journal articles on climate change impacts on work health, e.g., [44]	26 -> included 6
**Total sources**		**41 included from the initial 116 identified in the search**

**Table 2 ijerph-20-07004-t002:** Overview of some of the most cited articles on climate change impacts on WHS globally (as of 17 April 2023).

Authors	Cites	Title	Themes
Kjellstrom et al. (2016) [9]	423	Heat, human performance, and occupational health: A key issue for the assessment of global climate change impacts	heat, work, climate change, WHS
Xiang et al. (2014) [10]	376	Health impacts of workplace heat exposure: An epidemiological review	heat, WHS
Zander et al. (2015) [7]	360	Heat stress causes substantial labour productivity loss in Australia	heat, work, Australia
Ebi et al. (2021) [21]	243	Hot weather and heat extremes: Health risks	heat, health
Rowlinson et al. (2014) [51]	179	Management of climatic heat stress risk in construction: A review of practices, methodologies, and future research	heat, climate change, WHS
Varghese et al. (2018) [28]	126	Are workers at risk of occupational injuries due to heat exposure? A comprehensive literature review	WHS, heat
Hyatt et al. (2010) [11]	113	Regional maps of occupational heat exposure: Past, present, and potential future	work, heat

**Table 3 ijerph-20-07004-t003:** Number of included sources (see Appendix B) that reported on each type of climate-related hazard by Australian geographic category (urban/rural/Australia in general). The grey shading indicates high research coverage.

Climate-Related Hazards	Urban Australia	Rural/Regional Australia	Australia in General
Increased ambient Temperatures *	**16 sample sources (SS)** [9,12,16,18,20,21,23,28,47,48,54,55,56,57,58,59]	**6 SS** [4,23,51,54,60,61]	**32 SS** [4,7,9,10,11,12,17,18,19,20,21,22,23,27,28,44,45,46,47,48,50,51,54,55,56,57,58,59,60,61,62,63]
Extreme weather *	**5 SS** [20,23,47,54,58]	**6 SS** [4,23,31,54,61,64]	**18 SS** [4,11,17,20,23,24,27,31,44,47,50,54,58,61,64,65,66,67]
Air pollution *	**2 SS** [20,21]	**1 SS** [61]	**5 SS** [20,21,24,61,66]
Ultraviolet radiation *	**1 SS** [56]	**1 SS** [49]	**4 SS** [24,27,49,56]
Vector-borne disease and other *	**1 SS** [20]	**2 SS** [4,31]	**8 SS** [4,20,24,27,31,44,66,67]
Industrial transitions and emerging industries *	**0 SS**	**0 SS**	**1 SS** [24]
Changes in the built environment *	**4 SS** [20,21,47,48]	**0 SS**	**6 SS** [20,21,24,47,48,62]
Cascading climate change-related disasters **	**1 SS** [56]	**1 SS** [64]	**5 SS** [50,56,64,66,67]

* The seven climate-related hazards marked with an asterisk (*) were derived from the literature [49,50] and thus are ‘a priori’ themes. ** The a priori category marked with a double asterisk (**) was derived from the literature [30].

**Table 4 ijerph-20-07004-t004:** Catalogue of topics pertaining to interrelated climate change impacts on WHS (* Schulte et al. [1]; ** Ingham et al. [30]; *** Australian Bureau of Statistics, ABS [32]; Australian and New Zealand Standard Industry Classification, ANZSIC). Bolded topics indicate those for which a reasonable amount of existing knowledge was evident in our review for the Australian WHS context. Unbolded topics are those for which it was evident from our review that research focused on the Australian WHS context has been limited or negligible, despite the topics being recognised as important in global research in this area.

Domain	Topics
Climate-related hazards	**Increased ambient temperature *** Extreme weather * Air pollution * Ultraviolet radiation * Vector-borne disease and other biological hazards * Industrial transitions and emerging industries * Changes in the built environment * Cascading impacts **
Occupations	Accommodation and Food Services *** Administrative and Support Services *** **Agriculture, Forestry, and Fishing ***** Arts and Recreation Services*** **Construction ***** Education and Training *** Electricity, Gas, Water, and Waste Services *** Financial and Insurance Services *** Healthcare and Social Assistance *** Information Media and Telecommunications *** Manufacturing *** Military *** Mining *** Other Services (e.g., Tourism, Firefighters, and Sports) Professional Scientific and Technical Services *** Public Administration and Safety *** Rental, Hiring, and Real Estate Services *** Transport, Postal, and Warehousing *** Wholesale and Retail Trade ***
Health impacts	**Physical health** Mental health
Socio-economic impacts	Health-reduced workforce **Performance and productivity** Workplace issues (micro level) Economy and society (macro level)

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
