# Peer review of "Impacts of Climate Change on Work Health and Safety in Australia: A Scoping Literature Review"

_ijerph, 2023, doi:10.3390/ijerph20217004_

Round 1
Reviewer 1 Report
Comments and Suggestions for Authors
This study aimed to identify climate change impacts on Work Health and Safety (WHS) in Australia and associated knowledge gaps by mapping coverage of climate hazards, occupations at risk, and health and socio-economic impacts. The topic is interesting. Some comments for the authors to improve the quality of the manuscript are below.
1. In the paper, the authors mention that traditional systematic reviews are not applicable to interdisciplinary topics and research questions, and do not provide a proper definition of what kind of systematic review methods belong to traditional systematic reviews. Also, when referring again to the use of scoping reviews, there is no more detailed description of what a scoping review is, and a more unified and focused description of scoping reviews is needed. These two points need to be added in more detail by the authors.
2. It is worth noting that the databases recommended by experts for searching are mentioned in the text, and the search data are shown in more detail in Table 2, but common electronic databases such as Web of Science, Google Scholar, etc. are not mentioned in the text. What are the reasons for not searching in these databases?
3. In 3.1, when summarizing and analyzing the results of the disaster in each section, most of the consequences of the disaster are mainly stated in words, without data analysis or graphical presentation, which is not visual enough. The same problem is also found in 3.2.
4. Although it is a scoping review, the article mainly summarizes the consequences and impacts of disasters in declarative statements and lacks key graphical displays and data integration, which makes it less interesting and intuitive overall. The article would have more scientific value if graphs and data were included in key areas such as hazard classification and regional classification.
Comments on the Quality of English LanguageMinor editing of English language required.
Author Response
Responses to Reviewer 1
“Impacts of Climate Change on Work Health and Safety in Australia: A Scoping Literature Review”
International Journal of Environmental Research and Public Health (IJERPH) – Manuscript ID ijerph-2486703
The authors thank Reviewer 1 for providing us with very helpful comments. Please find the responses to your comments and suggestions in the table below, with copies of amended text from the manuscript coloured in red.
|
Reviewer 1 |
|
|
Comments |
Responses |
|
This study aimed to identify climate change impacts on Work Health and Safety (WHS) in Australia and associated knowledge gaps by mapping coverage of climate hazards, occupations at risk, and health and socio-economic impacts. The topic is interesting. Some comments for the authors to improve the quality of the manuscript are below. |
Thank you for your comment. |
|
1. In the paper, the authors mention that traditional systematic reviews are not applicable to interdisciplinary topics and research questions, and do not provide a proper definition of what kind of systematic review methods belong to traditional systematic reviews. Also, when referring again to the use of scoping reviews, there is no more detailed description of what a scoping review is, and a more unified and focused description of scoping reviews is needed. These two points need to be added in more detail by the authors. |
The key point we have endeavoured to make in the manuscript is that traditional systematic reviews are necessarily narrow in focus, whereas scoping reviews are generally broader and enable mapping of existing knowledge/ evidence relating to broad and interdisciplinary topic areas, like the topic covered by our scoping review. We have now added additional commentary in the Methods section (p. 4), with key references, to address the reviewer’s concerns and better differentiate and define these types of reviews : 2.1 Approach: Using scoping literature review as a methodological approach Our literature review follows the methodological approach of a scoping literature review [35], one type of systematic review [36]. The systematic approach is arguably the most rigorous literature review method and uses clearly defined processes to elicit all evidence on a given topic [37]–[39]. Such detailed research leads to reliable results and minimises effects on findings of possible researcher bias [37], [40]. According to the Joanna Briggs Institute (JBI), systematic reviews include various kinds of review methods, such as systematic reviews of experiences or meaningfulness; effectiveness; text and opinion/policy; prevalence and incidence; costs of a certain intervention, process, or procedure; etiology and risk; mixed methods; or diagnostic test accuracy; and umbrella reviews; and scoping reviews [36]. The Preferred Reporting Items for Systematic Reviews (PRISMA) Statement was extended in 2018 to Scoping Reviews – the PRISMA-ScR [41]. Scoping reviews, also called ‘scoping studies’ [36], allow for the preliminary assessment of the potential size and scope of available research literature in a certain field and aim to identify the nature and extent of research evidence [38]. We decided to conduct a scoping review, as it uses a systematic approach and is well suited to investigate broad interdisciplinary topics and research questions [39], such as those relating to the impacts of climate change on WHS. In contrast to other literature reviews seeking to identify the most significant items in a certain research field (critical review), aiming at identifying component reviews whilst excluding primary studies (umbrella reviews), or determining searches by time constraints (rapid review), the scoping review is limited in its completeness by scope constraints [38]. Furthermore, our research aim matches one of the primary purposes of a scoping literature review, which is “[t]o identify research gaps in the existing literature” [35, p. 21] and while “[s]pecifically designed to identify gaps in the evidence base where no research has been conducted, the study may also summarize and disseminate research findings” (p. 21). As a scoping literature review does not usually assess the quality of the identified literature, the identified gaps may “not necessarily identify research gaps where the research itself is of poor quality.” [35, p. 22]. Therefore, our scoping review was designed to “’map’ relevant literature in the field of interest” [35, p. 20]. Our mapped catalogue of topics highlights covered and uncovered aspects (quantity) of climate change impacts on WHS in Australia and offers recommendations on where future research could be more meaningful (quality). |
|
2. It is worth noting that the databases recommended by experts for searching are mentioned in the text, and the search data are shown in more detail in Table 2, but common electronic databases such as Web of Science, Google Scholar, etc. are not mentioned in the text. What are the reasons for not searching in these databases? |
We focused our searches on key databases that cover health and medical research because the focus of our review was the impacts of climate change on workplace/worker health (in Australia). We note, for example, that while Web of Science has very broad coverage, within it, MEDLINE (which we searched) is the key component of Web of Science that very broadly indexes health and medical journals. For the same reason, we focused on other key health and medical literature databases rather than conducting searches in the much broader Google Scholar search engine, which generates potentially infinite numbers of listed articles based on closest matches to search terms. Typically, when searching Google Scholar, it is only the first 100 or 200 listed articles in the output that can be considered for inclusion in a review because beyond those, listed articles become more and more unrelated to the specific topic of interest. For health topics like the one our review was focused on, it is the health and medical databases like those we searched (including MEDLINE from Web of Science) that will be the optimal databases to search. |
|
3. In 3.1, when summarizing and analyzing the results of the disaster in each section, most of the consequences of the disaster are mainly stated in words, without data analysis or graphical presentation, which is not visual enough. The same problem is also found in 3.2. |
Thank you for this comment. The approach we have taken is consistent with the type of review we have conducted – namely, a scoping review – which is designed to ‘map’ and summarise knowledge/evidence as presented by the original authors, usually without additional analysis. In particular, we based our work on the PRISMA-ScR guidelines, being the “Preferred Reporting Items for Systematic reviews and Meta-Analyses extension for Scoping Reviews” ([35], p. 2), which suggests authors of scoping literature reviews should “[s]ummarize and/or present the charting results as they relate to the review questions and objectives ([35], p. 2). Therefore, in the text sections you have mentioned, we have provided a fairly detailed textual synthesis of some of the key findings from the available literature on the respective topics but we have not provided additional analysis beyond that synthesis or offered graphical presentations of the data, which is instead provided in the cited source literature. See added text page 4: The Preferred Reporting Items for Systematic Reviews (PRISMA) Statement was extended in 2018 to Scoping Reviews – the PRISMA-ScR [41]. To provide a graphical presentation and increase the visualisation of the results, we have provided a conceptual map (Figure 2) on page 8:
Figure 2. Conceptual map illustrating research coverage of climate change impacts on WHS in Australia and associated knowledge gaps. |
|
4. Although it is a scoping review, the article mainly summarizes the consequences and impacts of disasters in declarative statements and lacks key graphical displays and data integration, which makes it less interesting and intuitive overall. The article would have more scientific value if graphs and data were included in key areas such as hazard classification and regional classification. |
Noting our response above, which is again relevant here, we also note that the manuscript is already quite lengthy and complex, as it covers many important topics of relevance to the impacts of climate change on work health and safety (and productivity) in Australia. A range of necessary tables have been included to ‘map’ the existing research literature in the topic area, and we were therefore reluctant to add to the length of the manuscript by including additional content. However, to provide a graphical presentation and increase the visualisation of the results, we have provided a conceptual map (Figure 2) on page 8 (see previous comment) |
|
Comments on the Quality of English Language: Minor editing of English language required. |
Thank you. English is the first language of several of the authors and we have reviewed and edited the paper further, accordingly (see track-change version of the manuscript). |

Reviewer 2 Report
Comments and Suggestions for Authors
The article deals with important topics and is well written. I just have a few minor comments:
1) it is worth writing in the introduction why the research form was literature analysis and not another
2) it is worth attempting to pose a research question or indicate a gap
3) a research scheme should be prepared
Author Response
Responses to Reviewer 2
“Impacts of Climate Change on Work Health and Safety in Australia: A Scoping Literature Review”
International Journal of Environmental Research and Public Health (IJERPH) – Manuscript ID ijerph-2486703
The authors thank Reviewer 2 for providing us with very helpful comments. Please find the responses to your comments and suggestions in the table below, with copies of amended text from the manuscript coloured in red.
|
Reviewer 2 |
|
|
Comments |
Responses |
|
The article deals with important topics and is well written. I just have a few minor comments: |
Thank you for your appreciative comment. |
|
1) it is worth writing in the introduction why the research form was literature analysis and not another |
We have actually provided this information/justification at the start of the Methods section, where we have specified and justified the research design. |
|
2) it is worth attempting to pose a research question or indicate a gap |
We have established the knowledge gap our review addresses and stated the aim of the research, as well as the research question it answers, at the end of the Introduction section. |
|
3) a research scheme should be prepared |
Thank you. The final ‘thematic template’ developed through the conduct of the scoping review is provided in Appendix A of the manuscript and reflects both a priori and emerging themes identified in the scoping review. We believe this addresses the reviewer’s concern here. To provide a graphical presentation and increase the visualisation of the results, we have now also provided a conceptual map (Figure 2) on page 8 (see previous comment). |

Reviewer 3 Report
Comments and Suggestions for Authors
Comments on the paper “Impacts of climate change on work health and safety in Australia: A scoping literature review.”
This paper reports findings of a search of scholarly literature regarding climate change related to worker health and safety in Australia. The whole article is well-summarized in the Abstract.
The authors undertook this project to fill a gap in scientific literature about climate change in Australia. Their aim statement consisted of two parts, (1) provide an overview of and mapping of what is known and (2) identifying key gaps in knowledge.
The Methodology section is not well organized. It seems to mix analyses, procedures, and findings. Most papers in IJERPH and other scholarly journals use subsections in the Methods. Subsections help authors stay focused and helps readers understand how the study was conducted. The authors need to revise this section. My suggestion is to use three subsections: (1) Approach: what is a scoping review and how is it different from other types of literature reviews; (2) Procedures: the five methodological steps followed, including formulating the research question; (3) Analyses: thematic clusters, developing the map, etc. Perhaps use appropriate sub-subsections for the longer topics.
The Findings section points out highlights on topics broken down into four main subsections on climate-related hazards, climate change impacts on occupations and industries, climate change impacts on Australians, and Socio-economic impacts. Each section has sub-subsection that I found helpful. The one figure in this section seems sufficient, but the caption should be moved from above to below the figure. Two appendices are used to document some of the findings organized logically. There are numerous mentions of “mapping” that were, in my view, organized lists. A suggestion is for the authors to consider putting a graphic in the Findings section that would depict something like a conceptual map.
The lengthy Discussion section shares the views of the authors on the topic areas having considerable literature all the way to topics with very little. The finding that occupational heat stress has been the topic of many studies involving Australian workers and their workplaces, including literature on Australian workers’ compensation claims for heat-related disorders and lost productivity when the working conditions require more frequent breaks. The authors identify some climate change topics that have received much less study than effects of hotter working conditions, in particular socio-economic impacts.
The authors reached the conclusion that many climate-change topics have been insufficiently studied. This is supported by the review. A suggestion for the authors is to consider acknowledging that a gap in literature may be due to multiple reasons including: (a) other researchers don’t regard the topic as important, (b) funding agencies don’t think the topic is important, and (c) the topic is not understood to be feasible to study. Another comment for the authors is to think carefully about telling employers they “should’ do this or that. This kind of advice is derived from the substance of articles rather on counting and organizing the 41 articles.
Limitation of the project are identified and discussed. My conclusion is that the project and the paper achieved the aim stated—provide an overview of and mapping of what is known and identify key gaps in knowledge, although I would say they identified gaps in peer reviewed papers on topics relevant to climate change effects on worker health and safety. I am not aware of any relevant literature missing from the listed references.
Author Response
Responses to Reviewer 3
“Impacts of Climate Change on Work Health and Safety in Australia: A Scoping Literature Review”
International Journal of Environmental Research and Public Health (IJERPH) – Manuscript ID ijerph-2486703
The authors thank Reviewer 3 for providing us with very helpful comments. Please find the responses to your comments and suggestions in the table below, with copies of amended text from the manuscript coloured in red.
|
Reviewer 3 |
||||||||||||||||||||||||||||||||||||||||||||||||||||||||||||
|
Comments |
Responses |
|||||||||||||||||||||||||||||||||||||||||||||||||||||||||||
|
Comments on the paper “Impacts of climate change on work health and safety in Australia: A scoping literature review.” This paper reports findings of a search of scholarly literature regarding climate change related to worker health and safety in Australia. The whole article is well-summarized in the Abstract. |
Thank you for your supportive comments. |
|||||||||||||||||||||||||||||||||||||||||||||||||||||||||||
|
The authors undertook this project to fill a gap in scientific literature about climate change in Australia. Their aim statement consisted of two parts, (1) provide an overview of and mapping of what is known and (2) identifying key gaps in knowledge. |
We agree with your comments. |
|||||||||||||||||||||||||||||||||||||||||||||||||||||||||||
|
The Methodology section is not well organized. It seems to mix analyses, procedures, and findings. Most papers in IJERPH and other scholarly journals use subsections in the Methods. Subsections help authors stay focused and helps readers understand how the study was conducted. The authors need to revise this section. My suggestion is to use three subsections: (1) Approach: what is a scoping review and how is it different from other types of literature reviews; (2) Procedures: the five methodological steps followed, including formulating the research question; (3) Analyses: thematic clusters, developing the map, etc. Perhaps use appropriate sub-subsections for the longer topics. |
Thank you for these suggestions. We have restructured and amended the Methods section accordingly.
2. Methodology Conducting a literature review is a “more or less systematic way of collecting and synthesizing previous research” [34, p. 333]. A literature review can provide an overview of disparate and interdisciplinary research areas, help synthesise research outcomes, showcase evidence on a meta-level and uncover areas where more research is needed [34]. For example, extant literature reviews on climate change impacts on WHS have used a systematic approach [27], a narrative approach [24], a comprehensive approach [28], an epidemiological approach [10], or a global approach [26]. The research question was addressed using the well-established scoping literature review [35]. 2.1 Approach: Using scoping literature review as a methodological approach Our literature review follows the methodological approach of a scoping literature review [35], one type of systematic review [36]. The systematic approach is arguably the most rigorous literature review method and uses clearly defined processes to elicit all evidence on a given topic [37]–[39]. Such detailed research leads to reliable results and minimises effects on findings of possible researcher bias [37], [40]. According to the Joanna Briggs Institute (JBI), systematic reviews include various kinds of review methods, such as systematic reviews of experiences or meaningfulness; effectiveness; text and opinion/policy; prevalence and incidence; costs of a certain intervention, process, or procedure; etiology and risk; mixed methods; or diagnostic test accuracy; and umbrella reviews; and scoping reviews [36]. The Preferred Reporting Items for Systematic Reviews (PRISMA) Statement was extended in 2018 to Scoping Reviews – the PRISMA-ScR [41]. Scoping reviews, also called ‘scoping studies’ [36], allow for the preliminary assessment of the potential size and scope of available research literature in a certain field and aim to identify the nature and extent of research evidence [38]. We decided to conduct a scoping review, as it uses a systematic approach and is well suited to investigate broad interdisciplinary topics and research questions [39], such as those relating to the impacts of climate change on WHS. In contrast to other literature reviews seeking to identify the most significant items in a certain research field (critical review), aiming at identifying component reviews whilst excluding primary studies (umbrella reviews), or determining searches by time constraints (rapid review), the scoping review is limited in its completeness by scope constraints [38]. Furthermore, our research aim matches one of the primary purposes of a scoping literature review, which is “[t]o identify research gaps in the existing literature” [35, p. 21] and while “[s]pecifically designed to identify gaps in the evidence base where no research has been conducted, the study may also summarize and disseminate research findings” (p. 21). As a scoping literature review does not usually assess the quality of the identified literature, the identified gaps may “not necessarily identify research gaps where the research itself is of poor quality.” [35, p. 22]. Therefore, our scoping review was designed to “’map’ relevant literature in the field of interest” [35, p. 20]. Our mapped catalogue of topics highlights covered and uncovered aspects (quantity) of climate change impacts on WHS in Australia and offers recommendations on where future research could be more meaningful (quality). 2.2 Procedures: Applying five stages of conducting a scoping literature review In conducting the scoping review, we followed the five-step process proposed by Arksey and O’Malley [35] as a methodological framework: identifying the research question; identifying relevant studies; study selection; charting the data; collating and summarising and reporting the results. In the first stage of identifying the research question, and inspired by Arksey and O’Malley [35], who formulated their research question starting with “What is known from the existing literature about…” (p. 23), we created the research question: What is known from the existing literature about how climate change impacts WHS in Australia? Answering this research question would allow us to understand the current state of knowledge at the nexus between climate change and WHS in Australia and to identify where extant literature is accumulated and what areas have received little attention, forming a knowledge gap. In the second stage of identifying relevant studies, we sought and confirmed literature to be included in the review. Arksey and O’Malley [35] suggest searching for relevant sources, predominately using electronic databases and existing networks and conferences, to identify what databases are generally used and key journals in a specific research field. Similar to Lundgren et al. [42], who investigated the effects of heat stress due to climate change on WHS, and Moda et al. [24], who explored climate change impacts on outdoor workers’ safety, we used the PubMed database. Furthermore, based on expert advice, we also utilised the databases: Cinahl Plus/EBSCO Host; Emcare/Ovid; Medline ALL; and Health Collection (Informit). From networks and a regional conference, we gathered some advice on key articles which were obtained. The reference lists of these seminal sources were hand-searched to identify additional studies, and search terms used in the reviewed sample were considered. For example, a previous literature review [42] on the effects of climate change and associated heat stress on WHS used the search terms ‘heat stress’, ‘occupational heat exposure’, ‘occupational heat stress’, ‘occupational heat strain’, ‘heat in/at workplace’, ‘work in the heat’, and ‘occupational heat stress AND climate change’. Based on our research question and in order to identify research evidence relating to all types of climate change hazards [1], [2], we used the search term ‘climate change’ instead of ‘heat*’. Furthermore, we combined the main concepts from our research question to formulate a search string, which combined ‘climate change’ using the Boolean operator ‘AND’ with ‘workplace’, AND ‘health’, AND ‘Australia’. The search was constructed to identify these search terms if they were located in a relevant source’s title, abstract or body. Our inclusion and exclusion criteria were informed by Snyder [39], who categorised literature reviews into three categories: systematic, semi-systematic and integrative. Our scoping study corresponds to Snyder’s [39] ‘semi-systematic’ category, allowing contributors to identify ‘themes in the literature’ and put up a ‘research agenda’, which corresponds to the purpose of the review. However, Snyder [39] also notes that “there are many other forms of literature reviews, and elements from different approaches are often combined” (p. 334). Our review benefited from such an adaptation, involving, in addition to peer-reviewed journal articles, the integration of research reports not published in academic journals (grey literature) to incorporate evidence from various disciplinary silos, such as ‘climate change’, ‘health’ and ‘WHS’ “in which the research is disparate and interdisciplinary…to uncover areas in which more research is needed” [39, p. 333]. Consequently, we included in the scoping review mainly peer-reviewed articles (38) and some other selected research reports (3). To refine our database search, ensure the selection of literature for inclusion was robust, and further validate our search strategy, we also sought and considered advice from experts in climate change and WHS research fields. Their suggestions confirmed what we identified as seminal works based on citation numbers. This way, expert advice added to the thoroughness of the review and offered new insights, thus expanding the range of included sources. We further followed the quality assessment approach of Gebayew and colleagues [43], ensuring the selected scholarly works were valuable for research or practice. Further inclusion criteria were the English language to avoid translation costs and the mention of ‘Australia’ as a significant study setting. Furthermore, the included studies had to be relevant to answering the research question, and we also included previous literature reviews if they met other inclusion criteria. A preliminary electronic search yielded a manageable number of studies for potential inclusion. Therefore, we decided not to restrict the date range and to include both qualitative and quantitative studies. A full reading of each paper subsequently identified in the final search (116 in total) confirmed initial eligibility perceptions derived from preliminary scans of titles and abstracts, with particular attention to excluding articles that mentioned the key search terms without further discussing the topic of climate change impacts on WHS in Australia. Applying these eligibility criteria to the initial 116 search results led to 75 exclusions and the inclusion of 41 sources (Appendix B), which focused on a range of occupational populations. Table 1 lists the databases and other sources considered in the search, as well as the search terms employed, the number of articles initially identified from each source, and numbers included based on eligibility during the selection process described further below. The overwhelming majority of the resulting articles investigated topics relating to ‘heat’.
Table 1. Search strategy: databases, terms and outcomes.
Figure 1. Eligible articles per publication year (N=41). Highly cited works in the field of climate change impacts on WHS have generally focused on heat impacts, for example, [7], [9], [10] (Table 2). Furthermore, high-quality journals publishing seminal works related to climate change impacts and WHS are the International Journal of Environmental Research and Public Health (e.g., [19], [24], [45], [46]); Environmental Research (e.g., [12], [26]); Science of the Total Environment (e.g., [47], [48]); and Policy and Practice in Health and Safety (e.g., [49], [50]). Table 2 lists some highly cited works on climate-change impacts, particularly heat impacts on WHS, according to citations.
Table 2. Overview of some of the most cited articles on climate change impacts on WHS, globally (as of 17 April 2023). 2.3 Analysis: Creating a final template Arksey and O’Malley’s [35] third and fourth stages describe the analysis process for the identified study sample. For the third stage of study selection, we used thematic template analysis [51] as a rigorous method to qualitatively analyse eligible source materials, the included 41 scholarly works. According to King and Brooks [51], thematic analysis is a technique that can be used to identify, analyse, synthesise and report recurring themes as general patterns in semi-structured literature reviews. Template analysis [51] consists of a six-step process to identify themes discussed in the extant literature on specific topics – in this review, climate change impacts on WHS in Australia. The six steps involve: familiarisation with the data, preliminary coding, clustering, producing an initial template, developing and applying the template, and writing up. We used Saldaña’s [52] thematic coding techniques and the QSR NVivo Version 12 (NVivo V12) software application to ensure rigorous organisation of the selected sources, systematic categorisation of the emerging themes, and a well-documented coding and analysis process. The NVivo V12 software application helped us organise the emerging themes meaningfully and retrieve coded data [52]. Eventually, a final template was developed through an iterative process by adding emerging themes, redefining existing themes, regrouping patterns, and merging similar themes. This final template (Appendix A) corresponds to Arksey and O’Malley’s [35] fourth stage of charting the data and lists the emerging themes, organised in interrelated clusters, as a basis for the later writing up. A cluster is a group of meaningfully ordered themes relating to each other, within a group or between them [51]. The final template provides an overview of what is known about climate change impacts on WHS in Australia and some trends in the research area. Our final template consists of four thematic clusters, embracing 62 themes and sub-themes. It formed the backbone for the last stage of the scoping literature review, which involved collating, summarising and reporting the results (Section 3). Methodological guidance suggests that, in this final stage of a scoping review, not all themes need to be discussed; instead, particular themes should be selected, for example, based on their frequency or ability to answer the research question [51]. Subsequently, the four main clusters identified in our review are discussed, to answer the research question, with these comprising: climate-related hazards impacting WHS in Australia (3.1); climate change impacts on occupations and industries in regional Australia (3.2); climate change impacts on health of Australian workers (3.3); and socio-economic impacts of climate change (3.4). The remaining clusters fall outside the scope of this review and may be further explored in future research. |
|||||||||||||||||||||||||||||||||||||||||||||||||||||||||||
|
The Findings section points out highlights on topics broken down into four main subsections on climate-related hazards, climate change impacts on occupations and industries, climate change impacts on Australians, and Socio-economic impacts. Each section has sub-subsection that I found helpful. The one figure in this section seems sufficient, but the caption should be moved from above to below the figure. Two appendices are used to document some of the findings organized logically. There are numerous mentions of “mapping” that were, in my view, organized lists. A suggestion is for the authors to consider putting a graphic in the Findings section that would depict something like a conceptual map. |
To provide a graphical presentation and increase the visualisation of the results, we have provided a conceptual map (Figure 2) on page 8:
Figure 2. Conceptual map illustrating research coverage of climate change impacts on WHS in Australia and associated knowledge gaps. Furthermore, we have moved the captions for all figures and tables as requested. |
|||||||||||||||||||||||||||||||||||||||||||||||||||||||||||
|
The lengthy Discussion section shares the views of the authors on the topic areas having considerable literature all the way to topics with very little. The finding that occupational heat stress has been the topic of many studies involving Australian workers and their workplaces, including literature on Australian workers’ compensation claims for heat-related disorders and lost productivity when the working conditions require more frequent breaks. The authors identify some climate change topics that have received much less study than effects of hotter working conditions, in particular socio-economic impacts. |
Thank you – we agree this encapsulated our discussion section. |
|||||||||||||||||||||||||||||||||||||||||||||||||||||||||||
|
The authors reached the conclusion that many climate-change topics have been insufficiently studied. This is supported by the review. A suggestion for the authors is to consider acknowledging that a gap in literature may be due to multiple reasons including: (a) other researchers don’t regard the topic as important, (b) funding agencies don’t think the topic is important, and (c) the topic is not understood to be feasible to study. Another comment for the authors is to think carefully about telling employers they “should’ do this or that. This kind of advice is derived from the substance of articles rather on counting and organizing the 41 articles. |
Thank you. We have added these points to the discussion and checked to ensure we are not telling employers what they should do, but rather simply conveying key information they may need (see pp. 25-26).
Globally, awareness is growing that climate change adaptation and mitigation strategies must be implemented to minimise future costs [26]. Examples include mitigating or avoiding occupational heat stress [74] and taking steps to limit the increase in average global temperature to less than 1.5°C [75]. In rural and regional Australia, communities are becoming more vulnerable to natural disasters caused by climate change; hence adaptive planning is needed to minimise damage, which requires a systematic analysis of the economic and social costs of climate change. Policies, guidelines, and heat management systems can also protect workers’ health, productivity, and income [11], [26]. Some Australian sectors (e.g., mining, construction and military) use heat stress risk assessments and prevention plans to support WHS and worker productivity [19]. From a practical point of view, the literature suggests strategies such as shifting work hours to cooler times of the day or seasons, working in the shade, applying cooling systems, supplying plenty of fresh water, and offering cool locations for rest breaks [9], [19], [26], while allowing work and rest patterns to be self-paced so workers’ bodies can maintain an optimum work capacity [9], [10], [46]. However, the current literature review also identified that measures like these that support workers’ health come with a cost, particularly for the companies. Hence, there is competition between workers’ health protection, investing in mitigation measures, and maintaining productivity in the work environment [11]. This scoping literature review contributes to practice in various ways. In terms of practical contributions, the ‘catalogue’ of topics provided in Table 4 can be used by various WHS stakeholders. Policymakers can use the catalogue as a source of information when developing WHS policies. In this way, the catalogue provides under-researched occupations with some visibility. Organisations may also use it to guide discussions about climate change and WHS with their stakeholders, particularly in geographical areas and occupations at risk. The catalogue illustrates the various elements (hazards; occupations; health; productivity) that should be considered when responding to the impacts of climate change on WHS, which are interrelated. As such, it may help organisations develop a holistic view, which acknowledges the multifaceted nature of climate impacts on WHS and enables them to provide adequate measures and support for their leaders and workers. This, in turn, has the potential to move organisations forward in becoming safe and sustainable workplaces when facing climate change. |
|||||||||||||||||||||||||||||||||||||||||||||||||||||||||||
|
Limitation of the project are identified and discussed. My conclusion is that the project and the paper achieved the aim stated—provide an overview of and mapping of what is known and identify key gaps in knowledge, although I would say they identified gaps in peer reviewed papers on topics relevant to climate change effects on worker health and safety. I am not aware of any relevant literature missing from the listed references. |
Thank you. We similarly believe we have captured all relevant articles for inclusion in the review. |
|||||||||||||||||||||||||||||||||||||||||||||||||||||||||||

Round 2
Reviewer 1 Report
Comments and Suggestions for Authors
The authors did a good job in addressing my comments to improve their manuscript. I have no further comments.